# Transformers are almost optimal metalearners for linear classification

**Roey Magen**
Weizmann Institute of Science
`roey.magen@weizmann.ac.il`

**Gal Vardi**
Weizmann Institute of Science
`gal.vardi@weizmann.ac.il`

## Abstract

Transformers have demonstrated impressive in-context learning (ICL) capabilities, raising the question of whether they can serve as metalearners that adapt to new tasks using only a small number of in-context examples, without any further training. While recent theoretical work has studied transformers' ability to perform ICL, most of these analyses do not address the formal metalearning setting, where the objective is to solve a collection of related tasks more efficiently than would be possible by solving each task individually. In this paper, we provide the first theoretical analysis showing that a simplified transformer architecture trained via gradient descent can act as a near-optimal metalearner in a linear classification setting. We consider a natural family of tasks where each task corresponds to a class-conditional Gaussian mixture model, with the mean vectors lying in a shared $k$-dimensional subspace of $\mathbb{R}^d$. After training on a sufficient number of such tasks, we show that the transformer can generalize to a new task using only $\widetilde{O}(k/\widetilde{R}^4)$ in-context examples, where $\widetilde{R}$ denotes the signal strength at test time. This performance (almost) matches that of an optimal learner that knows exactly the shared subspace and significantly outperforms any learner that only has access to the in-context data, which requires $\Omega(d/\widetilde{R}^4)$ examples to generalize. Importantly, our bounds on the number of training tasks and examples per task needed to achieve this result are independent of the ambient dimension $d$.

## 1 Introduction

Transformer-based models are the dominant architecture in both natural language processing (NLP) and computer vision. Since their introduction by Vaswani et al. [1], transformers have been scaled to produce remarkable advances in language modeling [2], image classification [3], and multimodal learning [4]. Their strength lies in their ability to model complex dependencies through attention mechanisms and to generalize across diverse tasks with minimal task-specific supervision.

One of the most intriguing emergent capabilities of large transformer models is *in-context learning* (ICL). In ICL, a model is given a short sequence of input-output pairs (called a *prompt*) from a particular (possibly new) task, and is asked to make predictions on test examples from that task without any explicit parameter updates. This ability to rapidly adapt to new tasks from a small number of examples, solely by conditioning on the prompt, has been observed in large language models [5], and is central to the ongoing shift toward prompt-based learning paradigms.

The ICL phenomenon is closely connected to the broader framework of *metalearning*, or "learning to learn" [6–8], which has been widely studied before. In metalearning, a learner is trained to perform well across a distribution of related tasks, thereby acquiring representations that allow for rapid adaptation to new tasks. It is often helpful to think of tasks as corresponding to individual users. For instance, generating personalized email completions. While each user provides limited task-specific data, such as writing style and personal preferences, there is a rich shared structure across users that

39th Conference on Neural Information Processing Systems (NeurIPS 2025).

can be exploited. Metalearning aims to leverage this structure to improve performance on each task beyond what would be possible if learned independently.

In this paper, we consider binary classification, where we model each task as a distribution $\mathcal{D}$ over labeled examples $(\boldsymbol{x}, y)$ in $\mathbb{R}^d \times \{\pm 1\}$. During training, the learner has access to $B$ datasets, where the $j$-th dataset consists of $N$ samples drawn i.i.d from task $\mathcal{D}_j$. We further assume that each task labeled by a classifier that relies on a common *representation* $h : \mathbb{R}^d \to \mathbb{R}^k$, where typically $k$ is smaller than the ambient dimension $d$. For each task $\mathcal{D}$, there is a classifier $f_{\mathcal{D}} : \mathbb{R}^k \to \{\pm 1\}$ such that $f_{\mathcal{D}} \circ h$ has high accuracy on $\mathcal{D}$. Our interest lies in studying families of tasks for which knowing this shared representation $h$ substantially reduces the number of samples required to learn each task separately.

**Metalearning Objective.** Assuming that the tasks $\mathcal{D}_1, \ldots, \mathcal{D}_B$ are themselves drawn i.i.d. from an unknown metadistribution $\Omega$, the goal is to output a representation $\hat{h}$ that can be specialized to a new unseen task $\mathcal{D} \sim \Omega$. In the modern view, the model is first trained on tasks $\mathcal{D}_1, \ldots, \mathcal{D}_B$, each task contain $N$ samples. Then, at test time, the learner is given $M$ in-context labeled samples drawn i.i.d. from some new task $\mathcal{D}$, and needs to classify a new sample from $\mathcal{D}$, without further training. To evaluate performance, mostly the required number of in-context labeled samples $M$ that required to ensure (with high probability) small error, we consider two benchmark baselines:

- **Single-task optimal learner:** An optimal algorithm, in terms of the number of required samples, that has access only to samples from the new task $\mathcal{D}$.
- **Optimal learner with access to the ground-truth representation:** An optimal algorithm that has access both to samples of the new task and to the ground-truth representation $h$.

The above discussion naturally motivates the study of transformers from the lens of metalearning. In particular, we are interested in understanding whether, and under what conditions, transformer architectures can act as (optimal) *metalearners*. Specifically,

*Can a transformer outperform a* single-task optimal learner *(that only have access to the in-context data), and potentially approach the performance of an* optimal learner with access to the ground-truth representation*?*

## 1.1 Our contribution

To address the question of whether transformers can serve as effective *metalearners*, we analyze their behavior in the well-studied Gaussian mixture framework. In this setting, each task is a random instance of a class-conditional Gaussian mixture model in $\mathbb{R}^d$, with identical spherical covariance and opposite means. The task-specific signal vector $\boldsymbol{\mu}$ is sampled from a shared low-dimensional subspace of dimension $k \leq d$. Formally, for some semi-orthogonal matrix $\boldsymbol{P} \in \mathbb{R}^{d \times k}$ and signal strength $R > 0$, each task $\tau = 1, \ldots, B$ is defined by:

$$\boldsymbol{\mu}_\tau \overset{\text{i.i.d.}}{\sim} \boldsymbol{P} \cdot \text{Unif}(R \cdot \mathbb{S}^{k-1}), \quad y_{\tau,i} \overset{\text{i.i.d.}}{\sim} \text{Unif}(\{\pm 1\}), \quad \boldsymbol{z}_{\tau,i} \overset{\text{i.i.d.}}{\sim} \mathsf{N}(\boldsymbol{0}, \boldsymbol{I}_d), \quad \boldsymbol{x}_{\tau,i} = y_{\tau,i} \boldsymbol{\mu}_\tau + \boldsymbol{z}_{\tau_i}.$$

At test time, in-context examples are also drawn from a class-conditional Gaussian mixture model, but potentially with a different signal-to-noise ratio determined by a test-time signal strength $\tilde{R}$. That is, the meta-distribution may shift between training and testing. We consider a simplified transformer with linear attention, a setup similar to many prior works on ICL [9–12]. The transformer is trained via gradient descent (GD) on the logistic or exponential loss over the above random linear classification tasks.

Our main contributions are as follows:

- We prove that if the transformer is trained on a sufficiently large number of tasks, then the number of in-context samples required at test time to achieve a small constant error on a new task (without parameters update) is $\tilde{O}(k/\tilde{R}^4)$, where $\tilde{R}$ denote the test-time signal strength. In contrast, any single-task learner, which has access only to samples from the new task, require at least $\Omega(d/\tilde{R}^4)$ samples (see our discussion in Remark 2.1 on the information-theoretic lower bound by Giraud and Verzelen [13]). To our knowledge, this is the first theoretical result establishing such a guarantee for metalearning with transformers in a linear classification setting.

- Using the lower bound from Giraud and Verzelen [13], we show that even an *optimal learner with access to the ground-truth representation*, one that knows the true shared low-dimensional subspace $\boldsymbol{P}$, requires at least $\Omega(k/\tilde{R}^4)$ samples to achieve a small constant error. This implies that transformers, in our setting, are nearly optimal metalearners for linear classification.

- Our analysis also yields a characterization of the number of pretraining tasks required to generalize effectively at test time. Specifically, we derive an explicit relationship between the number of tasks $B$ and the signal strength $R$, which controls the signal-to-noise ratio (SNR) during training. We find that it is sufficient to train the transformer on $B = (k/\text{SNR}^2)$ tasks and $N = O(1/\text{SNR}^2) \vee 1$ samples per task, to (almost) match the performance of an optimal metalearner, without any dependence on the ambient dimension $d$.

- Finally, while Frei and Vardi [11] analyze a similar setting without assuming a shared representation (i.e., they assumed $k = d$) and require a strong assumption $R = \Omega(\sqrt{d})$, we show that it suffices to have $R = \tilde{\Omega}(1)$ for achieving in-context generalization, even when $k = d$. We note that a single-task optimal learner needs only $O(1)$ samples when the signal strength already equals $\Theta(d^{1/4})$, whereas it is information-theoretically impossible to achieve small error when the signal strength is $o(1)$, regardless of how many samples are available (see again Remark 2.1). Thus, learning a Gaussian mixture is challenging in the regime where the signal strength is between $\Omega(1)$ and $O(d^{1/4})$, and we cover the case where both $R$ and $\tilde{R}$ are in this regime.

## 1.2 Related Work

**In context learning.** Following the initial experiments of Garg et al. [14], which demonstrated empirically that transformers can perform in-context learning of various function classes, such as linear functions, two-layer neural networks, and decision trees, a number of works have explored what types of algorithms are implemented by trained transformers. Akyürek et al. [15], Bai et al. [16], Von Oswald et al. [17] provided expressivity results showing that transformers can implement a wide range of in-context algorithms such as least squares, ridge regression, Lasso and gradient descent on two-layer neural networks. Wies et al. [18] provided a PAC framework for in-context learnability, and established finite sample complexity guarantees. Huang et al. [19] investigated the training dynamics of a one-layer transformer with softmax attention trained by GD in a regression setting. Focusing on linear regression, Wu et al. [10] established a statistical task complexity bound. Ahn et al. [20] and Mahankali et al. [21] demonstrated that a one-layer transformer minimizing the pre-training loss effectively implements a single step of gradient descent. Zhang et al. [22] additionally developed guarantees for the convergence of (non-convex) gradient flow dynamics.

In the linear *classification* setting, Shen et al. [12] showed that a linear transformer trained via gradient descent is equivalent to the optimal logistic regressor, whenever the number of training tasks $B$, the number of samples per task $N$, and the test prompt length $M$, are all tend to infinity. Li et al. [23] showed that a single-layer linear attention model can learn the optimal binary classifier under the squared loss, with a focus on semi-supervised learning. The work most closely related to ours is Frei and Vardi [11], who studied the behavior of linear transformers via an analysis of the implicit regularization of gradient descent (similar to our approach). As we already mentioned, they analyzed a setting with a strong signal $R = \Omega(\sqrt{d})$ while we allow $R = \tilde{\Omega}(1)$. Moreover, we emphasize that none of the above papers addresses metalearning in the sense studied in our work.

**Metalearning.** There is a large body of research on metalearning, often associated with related concepts or alternative names such as multitask learning, transfer learning, learning to learn, and few-shot learning (See Thrun and Pratt [24] for an early overview). Baxter [6] provided distribution-free sample complexity bounds for metalearning. A long line of works [25–30] has developed computationally efficient metalearning algorithms, such as MAML (Model-Agnostic Meta-Learning) under various settings, primarily for regression tasks. Several works consider metalearning with a shared low-dimensional linear representation, which resembles our setting, albeit their metalearners are not related to transformers [31–33, 7, 34, 29]. To our knowledge, the only work that explores a form of metalearning in transformers is Oko et al. [35], which studies a linear transformer architecture augmented with a nonlinear MLP layer. For target functions of the form $f(\boldsymbol{x}) = \sigma(\langle \boldsymbol{\mu}, \boldsymbol{x} \rangle)$, where $\boldsymbol{\mu} \in \mathbb{R}^d$ lies in a $k$-dimensional subspace, they show that the model can learn in-context with a

prompt length that scales only with $k$. The key differences from our work are: First, their focus is on regression rather than classification. Second, they employ a somewhat artificial two-step optimization procedure – first applying gradient descent on the MLP layer, and only afterwards performing empirical risk minimization (ERM) on the attention layer. In contrast, we consider standard end-to-end gradient descent. Third, they require that the number of tasks and samples during training scales with the ambient dimension $d$.

**Implicit Regularization in Transformers.** Our theoretical analysis begins by examining the *implicit regularization* induced by gradient descent in transformer models. We refer readers to the survey by Vardi [36] for a broader overview. The convex linear transformer architecture we study is linear in the vectorized parameters, which, following Soudry et al. [37], implies that gradient descent converges in direction to the max-margin classifier. More general transformer architectures are non-convex, but many subclasses exhibit parameter homogeneity and thus converge (in direction) to Karush-Kuhn-Tucker (KKT) points of max-margin solutions [38, 39]. Another line of work investigates the implicit bias of gradient descent in softmax-based transformers [40–43], typically under stronger assumptions about the structure of the training data.

## 2 Preliminaries

**Notations.** We use bold-face letters to denote vectors and matrices, and let $[n]$ be shorthand for $\{1, 2, \ldots, n\}$. Let $\boldsymbol{I}_d$ be the $d \times d$ identity matrix, and let $\boldsymbol{0}_d$ (or just $\boldsymbol{0}$, if $d$ is clear from the context) denote the zero vector in $\mathbb{R}^d$. We let $\|\cdot\|$ denote the Euclidean norm. The Frobenius norm of a matrix is denoted $\|\boldsymbol{W}\|_F$. We use $a \vee b = \max(a, b)$ and $a \wedge b := \min(a, b)$. We use standard big-Oh notation, with $\Theta(\cdot), \Omega(\cdot), O(\cdot)$ hiding universal constants and $\tilde{\Theta}(\cdot), \tilde{\Omega}(\cdot), \tilde{O}(\cdot)$ hiding constants and factors that are polylogarithmic in the problem parameters.

### 2.1 Data Generation Setting

We consider the following metadistribution during training:

**Assumption 2.1** (training-time task distribution). *Fix some $k \leq d$ and let $\boldsymbol{P} \in \mathbb{R}^{d \times k}$ be a semi-orthogonal matrix, i.e. $\boldsymbol{P}^\top \boldsymbol{P} = \boldsymbol{I}_k$. Let $B, N \geq 1$ and signal strength $R > 0$. For any task $\tau \in [B]$, the input-label pairs $(\boldsymbol{x}_{\tau,i}, y_{\tau,i})_{i=1}^{N+1}$ in task $\tau$ satisfy the following:*

1. *Let $\boldsymbol{\mu}'_\tau$ be sampled i.i.d from the distribution $\mathsf{Unif}(R \cdot \mathbb{S}^{k-1})$, i.e., the uniform distribution on the sphere of radius $R$ in $k$ dimensions.*

2. *Set $\boldsymbol{\mu}_\tau = \boldsymbol{P}\boldsymbol{\mu}'_\tau$ to be the isometric embedding of $\boldsymbol{\mu}'_\tau$ in $\mathbb{R}^d$ under $\boldsymbol{P}$.*

3. *Let $\boldsymbol{z}_{\tau,i} \overset{i.i.d.}{\sim} \mathsf{N}(\boldsymbol{0}, \boldsymbol{I}_d)$, and $y_{\tau,i} \overset{i.i.d.}{\sim} \mathsf{Unif}(\{\pm 1\})$, where $\boldsymbol{\mu}_\tau$, $\boldsymbol{z}_{\tau,i}$ and $y_{\tau,i}$ are mutually independent.*

4. *Conditioned on the task parameter $\boldsymbol{\mu}_\tau$, set $\boldsymbol{x}_{\tau,i} := y_{\tau,i}\boldsymbol{\mu}_\tau + \boldsymbol{z}_{\tau,i}$.*

Thus, the above assumption states that each pretraining task is a class-conditional Gaussian mixture, with two opposite Gaussians, where the direction of the cluster means for each task is drawn randomly from a $k$-dimensional subspace in $\mathbb{R}^d$. Next, we introduce the test-time distribution, which may generalize the pretraining distribution by allowing in-context examples to have a different cluster mean size and sample size (denoted by $\tilde{R}$ and $M + 1$) than those during training ($R$ and $N + 1$).

**Assumption 2.2** (test-time task distribution). *Let $M \geq 1$ be the number of in-context examples and $\tilde{R} > 0$ be the signal strength during test time. The input-label pairs $(\boldsymbol{x}_i, y_i)_{i=1}^{M+1}$ in the test task satisfy the following:*

1. *Let $\boldsymbol{\mu}'$ be sampled from the distribution $\mathsf{Unif}(\tilde{R} \cdot \mathbb{S}^{k-1})$, i.e., the uniform distribution on the sphere of radius $\tilde{R}$ in $k$ dimensions.*

2. *Set $\boldsymbol{\mu} = \boldsymbol{P}\boldsymbol{\mu}'$, where $\boldsymbol{P}$ is from assumption 2.1, be the isometric embedding of $\boldsymbol{\mu}'$ in $\mathbb{R}^d$.*

3. *Let $\boldsymbol{z}_i \overset{i.i.d.}{\sim} \mathsf{N}(\boldsymbol{0}, \boldsymbol{I}_d)$, and $y_i \overset{i.i.d.}{\sim} \mathsf{Unif}(\{\pm 1\})$, where $\boldsymbol{\mu}$, $\boldsymbol{z}_i$ and $y_i$ are mutually independent.*

4. *Conditioned on the task parameter $\boldsymbol{\mu}$, set $\boldsymbol{x}_i := y_i\boldsymbol{\mu} + \boldsymbol{z}_i$.*

Since the cluster means in our training and test distributions have norms $R$ and $\tilde{R}$, and the deviation from the cluster means has a standard Gaussian distribution and hence norm of roughly $\sqrt{d}$, we call the ratios $\frac{R}{\sqrt{d}}$ and $\frac{\tilde{R}}{\sqrt{d}}$ the *signal-to-noise ratios* (SNR for short).

## 2.2 Attention Model & Tokenization

In our setting, each example is a task that consists of a sequence of $(\boldsymbol{x}, y)$ pairs. In order to encode such a task and provide it as an input to the transformer (i.e., to tokenize it), we use the following embedding matrix:

$$\boldsymbol{E} = \begin{pmatrix} \boldsymbol{x}_1 & \boldsymbol{x}_2 & \cdots & \boldsymbol{x}_N & \boldsymbol{x}_{N+1} \\ y_1 & y_2 & \cdots & y_N & 0 \end{pmatrix} \in \mathbb{R}^{(d+1)\times(N+1)}. \tag{1}$$

That is, each of the $N+1$ examples is given in a separate column, and for the $(N+1)$-th example we do not encode the label and place $0$ instead. The single-head transformer with softmax attention [1] is parametrized by query, key, and value matrices: $\boldsymbol{W}^V \in \mathbb{R}^{d_e \times d_e}$, $\boldsymbol{W}^K, \boldsymbol{W}^Q \in \mathbb{R}^{d_k \times d_e}$. Then, softmax attention is defined by

$$f(\boldsymbol{E}; \boldsymbol{W}^K, \boldsymbol{W}^Q, \boldsymbol{W}^V) = \boldsymbol{E} + \boldsymbol{W}^V \boldsymbol{E} \cdot \mathrm{softmax}\left(\frac{(\boldsymbol{W}^K \boldsymbol{E})^\top \boldsymbol{W}^Q \boldsymbol{E}}{\rho}\right), \tag{2}$$

where $\rho$ is a fixed normalization that may depend on $N$ and $d_e$, but is not learned. We focus on *linear transformers*, where the softmax is replaced with the identity function. Following many prior works (e.g., Von Oswald et al. [17], Zhang et al. [22], Ahn et al. [20]), we consider a parameterization where the key and query matrices $\boldsymbol{W}^K, \boldsymbol{W}^Q$ are merged into $\boldsymbol{W}^{KQ} := (\boldsymbol{W}^K)^\top \boldsymbol{W}^Q$. Our objective is to use the first $N$ columns of Eq. 1 to predict $\boldsymbol{x}_{N+1}$. Similar to prior works, we consider a convex parameterization of the linear transformer, obtained by fixing some of the parameters to $0$ or $1$ (see details in Appendix A), which results in the following prediction for the label of $\boldsymbol{x}_{N+1}$:

$$\hat{y}(\boldsymbol{E}; \boldsymbol{W}) = \left(\frac{1}{N}\sum_{i=1}^{N} y_i \boldsymbol{x}_i\right)^\top \boldsymbol{W} \boldsymbol{x}_{N+1}. \tag{3}$$

Here, the trained parameters are $\boldsymbol{W}$. We note that the model in Eq. 3 has become a common toy model for analyzing in-context learning both for both regression [10, 9] and classification [11, 12].

## 2.3 Gradient Descent & Implicit Bias

Given a task $\tau$, we define the embedding matrix $\boldsymbol{E}_\tau$ using the labeled examples $(\boldsymbol{x}_{\tau,i}, y_{\tau i})_{i=1}^{N+1}$ from Assumption 2.1, tokenized according to Eq. 1. We consider linear transformers (Eq. 3) trained to minimize the prediction loss on the final token $\boldsymbol{x}_{\tau,N+1}$, with ground-true label $y_{\tau,N+1}$. Formally, for a training dataset $\{(\boldsymbol{E}_\tau, y_{\tau,N+1})\}_{\tau=1}^{B}$, we define the empirical loss:

$$\mathcal{L}(\boldsymbol{W}) := \frac{1}{B}\sum_{\tau=1}^{B} \ell\big(y_{\tau,N+1} \cdot \hat{y}(\boldsymbol{E}_\tau; \boldsymbol{W})\big),$$

where $\hat{y}(E; W)$ is the prediction function from Eq. 3 and $\ell$ is either the logistic loss $\ell(z) = \log(1 + \exp(-z))$ or the exponential loss $\ell(z) = \exp(-z)$. We train on this objective using gradient descent: $\boldsymbol{W}_{t+1} = \boldsymbol{W}_t - \alpha\nabla\mathcal{L}(\boldsymbol{W}_t)$, where $\alpha > 0$ is a fixed learning rate. Since $\hat{y}(\boldsymbol{E}_\tau; \boldsymbol{W})$ is linear in $\boldsymbol{W}$, gradient descent has an implicit bias towards maximum-margin solutions, as formalized in the following theorem:

**Theorem 2.3** (Soudry et al. [37]). *Let $\boldsymbol{W}_{MM}$ denote the solution to the max-margin problem:*

$$\boldsymbol{W}_{MM} := \operatorname*{arg\,min}_{\boldsymbol{U}} \|\boldsymbol{U}\|_F^2 \text{ s.t. } \left(1/N \sum_{i=1}^{N} y_{\tau,i}\boldsymbol{x}_{\tau,i}\right)^\top \boldsymbol{U} y_{\tau,N+1}\boldsymbol{x}_{\tau,N+1} \geq 1, \forall \tau = 1, \ldots, B. \tag{4}$$

*If the above problem is feasible and the learning rate $\alpha$ is sufficiently small, then $\boldsymbol{W}_t$ converges in direction to $\boldsymbol{W}_{MM}$, that is $\boldsymbol{W}_t/\|\boldsymbol{W}_t\| \to c\boldsymbol{W}_{MM}$, as $t \to \infty$, for some constant $c > 0$.*

Thus, the max-margin solution $\boldsymbol{W}_{\mathrm{MM}}$ characterizes the asymptotic behavior of gradient descent for our model. In the remainder of this work, we analyze the ability of this max-margin solution to perform in-context learning and metalearning.

### 2.4 Information-Theoretic Lower Bounds for Single-Task and Metalearners

We begin by characterizing the minimax test error for algorithms that only have access to the $M$ labeled examples from the test task $(\boldsymbol{x}_1, y_1), \ldots, (\boldsymbol{x}_M, y_M)$, but not to the underlying subspace $\boldsymbol{P}$, and need to predict the label of $\boldsymbol{x}_{M+1}$. This models the performance of an *optimal single-task learner*. Importantly, this lower bound holds even when $\boldsymbol{P}$, as defined in Assumption 2.1, is drawn uniformly at random from the space of all semi-orthogonal matrices in $\mathbb{R}^{d \times k}$. We emphasize that our main generalization result (Theorem 3.1) applies in the worst-case setting—i.e., it holds for any fixed $\boldsymbol{P}$. However, when $\boldsymbol{P}$ is sampled uniformly at random, the induced task mean $\boldsymbol{\mu}$ becomes uniformly distributed on the sphere of radius $\tilde{R}$ in $\mathbb{R}^d$. In this setting, we recover the following lower bound:

**Remark 2.1** (Giraud and Verzelen [13], Appendix B). *The minimax test error for Gaussian classification with identical spherical covariance and opposite means, as defined in Assumption 2.2 with $k = d$, is at least $c \cdot \exp\left(-c' \cdot \min\left\{R^2, \frac{MR^4}{d}\right\}\right)$, for some absolute constants $c, c' > 0$. In particular, when $R = \Omega(1)$, the number of samples required to achieve small constant error must satisfy $M = \Omega(d/R^4)$. While for $\tilde{R} = o(1)$ it is impossible to learn with small error.*

We now consider the setting where the learner is granted full access to the underlying subspace $\boldsymbol{P}$, in addition to the in-context labeled examples and the test point $(\boldsymbol{x}_1, y_1), \ldots, (\boldsymbol{x}_M, y_M), \boldsymbol{x}_{M+1}$. This models the performance of an *optimal learner that knows the shared representation*.

**Remark 2.2.** *Consider the same Gaussian classification model, where the mean vectors lie in a low-dimensional subspace $\boldsymbol{P} \subseteq \mathbb{R}^d$, as described in Assumption 2.2 with $k \leq d$. Then, the minimax test error for any algorithm with access to $\boldsymbol{P}$ is at least $c \cdot \exp\left(-c' \cdot \min\left\{R^2, \frac{MR^4}{k}\right\}\right)$, for some absolute constants $c, c' > 0$.*

The proof of this remark follows by a direct reduction to the bounds established by Giraud and Verzelen [13] and is included in the appendix for completeness.

## 3 Main Result

We make the following assumptions:

**Assumption A.** *Let $\delta > 0$ be a desired probability of failure. There exists a sufficiently large universal constant $C$ (independent in $d, B, k, N$ and $\delta$), such that the following conditions hold:*

*(A1) The signal strength $R$ satisfies: $C \log(B/\delta) \leq R^2 \leq \frac{d}{C \log^2(B/\delta)}$*

*(A2) Dimension $d$ should be sufficiently large: $d \geq C \log^4(B/\delta)$.*

*(A3) Number of samples per task $N$ should be sufficiently large: $N \geq C(d/R^2) \vee 1$*

Assumption (A1) provides explicit bounds on the signal-to-noise ratio (SNR). We emphasize that when $R = o(1)$, learning with small error is information-theoretically impossible (see Remark 2.1). In contrast, when $R = \Omega(\sqrt{d})$, the learning problem is solvable even by a trained transformer that observes only a single example per task, i.e., $N = M = 1$ (see Frei and Vardi [11]). Assumptions (A2) and (A3) are technical conditions introduced to guarantee certain concentration inequalities involving sub-exponential random variables.

Recall that the transformer with parameters $\boldsymbol{W}$ makes predictions by embedding the set $\{(\boldsymbol{x}_i, y_i)\}_{i=1}^{M} \cup \{(\boldsymbol{x}_{M+1}, 0)\}$ into a matrix $\boldsymbol{E}$ (as in Eq. 1), and then predicting $y_{M+1}$ as $\text{sign}(\hat{y}(\boldsymbol{E}; \boldsymbol{W}))$. Our objective is to characterize the *expected risk* of the max-margin solution, namely, the probability that the transformer misclassifies the test example $(\boldsymbol{x}_{M+1}, y_{M+1})$ when parameterized by $\boldsymbol{W}_{mm}$.

We now state our main result, which shows that transformers can serve as near-optimal metalearners:

**Theorem 3.1.** *Let $\delta \in (0, 1)$ be arbitrary. There are absolute constants $C > 1, c > 0$ such that if Assumption A holds (w.r.t. $C$), then with probability at least $1 - \delta$ over the draws of $\{\boldsymbol{\mu}_\tau, (\boldsymbol{x}_{\tau,i}, y_{\tau,i})_{i=1}^{N+1}\}_{\tau=1}^{B}$, when sampling a new task $\{\boldsymbol{\mu}, (\boldsymbol{x}_i, y_i)_{i=1}^{M+1}\}$, the max-margin solution*

*from Eq. 4 satisfies*

$$\mathbb{P}_{(\boldsymbol{x}_i,y_i)_{i=1}^{M+1},\,\boldsymbol{\mu}}\big(\operatorname{sign}(\hat{y}(\boldsymbol{E};\boldsymbol{W}_{MM})) \neq y_{M+1}\big)$$

$$\leq 6\exp\left(-\frac{c}{\log^2(B/\delta)}\cdot\left(1\wedge\sqrt{\frac{BR^2}{dk}}\right)\cdot\left(\sqrt{k}\wedge\tilde{R}\wedge\sqrt{\frac{M\tilde{R}^4}{k}}\right)\right)$$

Let us make a few observations on the above theorem:

- Assume that the number of tasks $B$ during training is sufficiently large, specifically $B = \Omega(dk/R^2)$, so that the term $\left(1\wedge\sqrt{\frac{BR^2}{dk}}\right) = \Theta(1)$. To achieve an arbitrarily small constant test error (e.g., at most 0.001), it suffices to have $k = \tilde{O}(1), \tilde{R} = \tilde{O}(1)$ and $M = \tilde{O}(k/\tilde{R}^4)$ in-context examples. By Remark 2.1, an optimal algorithm that only has access to the samples from a new task needs $\Omega(d/\tilde{R}^4)$ samples to achieve a small constant error. Therefore, a trained transformer can learn the small subspace during training and enjoy a better in-context sample complexity than such an optimal algorithm whenever $k \ll d$. In particular, if $k \leq d^\alpha$, for some $\alpha < 1$, we obtain a polynomial improvement in the sample complexity whenever $\tilde{R} = o(d^{1/4})$.

- In fact, a trained transformer is almost an optimal metalearner: given enough tasks during training, it suffices to have $M = \tilde{O}(k/\tilde{R}^4)$ in-context examples to achieve constant error. This matches the lower bound for an optimal learning algorithm that has full access to the subspace $\boldsymbol{P}$ (See Remark 2.2). To achieve an error at most $\epsilon$ (with probability at least $1 - \delta$ over the training data), the trained transformer will need $O\left((k/\tilde{R}^4)\cdot\log^2(1/\epsilon)\cdot\log^4(B/\delta)\right)$ in-context samples, while the lower bound for optimal learner that has access to the ground truth subspace is $\Omega\left((k/\tilde{R}^4)\cdot\log(1/\epsilon)\right)$ samples.

- A common assumption in the metalearning literature is that the training and test tasks are drawn from the same metadistribution. In our setting, this corresponds to the case where $R = \tilde{R}$. Under this assumption, our analysis shows that a trained transformer can generalize as long as $R = \tilde{\Omega}(1)$, $k = \tilde{\Omega}(1)$, and the number of pertaining tasks $B$ and in-context samples $M$ are sufficiently large. This improves upon the result of Frei and Vardi [11], which required the stronger condition of $R = \tilde{\Omega}(d^{1/2})$. Note that when $R$ is at least $\Omega(d^{1/4})$ a single-task optimal learner needs only $O(1)$ samples, and for $R = o(1)$ learning is impossible by Remark 2.1. Hence, our weaker condition on $R$ allows us to cover the challenging regime where $\Omega(1) \leq R \leq O(d^{1/4})$.

- Moreover, our analysis yields a tighter dependence on the number of training tasks $B$ required for generalization in the high signal regime compared to Frei and Vardi [11]. As a concrete example, suppose $R = \tilde{R} = \tilde{\Theta}(\sqrt{d})$, $M = \Theta(1)$, and $d = k$. Then our analysis shows that it suffices to train on $B = \tilde{O}(1)$ tasks, whereas the result of Frei and Vardi [11] implies that in this case $B$ should be $\tilde{O}(\sqrt{d})$.

- At first glance, it may seem that the number of tasks $B$ and the number of samples per task during training $N$ must scale with the ambient dimension $d$. This impression arises because the noise terms in the data (i.e., $\boldsymbol{z}_{\tau,i}$ from Assumption 2.1) have norm $\|\boldsymbol{z}_{\tau,i}\| \simeq \sqrt{d}$.[1] Consequently, it is natural to ask how $B$ and $N$ relate to the signal-to-noise ratio (SNR) during training, which is defined as $\|\boldsymbol{\mu}_{\tau,i}\| / \|\boldsymbol{z}_{\tau,i}\| \simeq R/\sqrt{d}$. More importantly, can we eliminate any dependence of $B$ and $N$ on $d$, which may be very large? Perhaps surprisingly, the answer is positive. By substituting $R^2 = d \cdot \mathrm{SNR}^2$ into Theorem 3.1 and Assumption (A3), we find that it suffices to train the transformer on $B = O(k/\mathrm{SNR}^2)$ tasks and $N = O(1/\mathrm{SNR}^2) \vee 1$ samples per task, to achieve performance equivalent to an optimal metalearner, without any dependence on $d$.

---

[1] We emphasize that this assumption is without loss of generality. Indeed, if $\boldsymbol{z}_{\tau,i} \sim \mathsf{N}(\boldsymbol{0}, \sigma^2\boldsymbol{I})$ for some $\sigma \neq 1$, we can rescale $R$ by a factor of $\sigma$ and plug it into our results, since the dynamics of GD remain the same.

## 4 Proof Sketch

Let $\boldsymbol{W} := \boldsymbol{W}_{\mathrm{MM}}$ be the max-margin solution (Eq. 4), i.e.

$$\boldsymbol{W} := \arg\min\{\|\boldsymbol{U}\|_F^2 : \hat{\boldsymbol{\mu}}_\tau^\top \boldsymbol{U} y_\tau \boldsymbol{x}_\tau \geq 1, \ \forall \tau = 1, \ldots, B\}. \tag{5}$$

where $\hat{\boldsymbol{\mu}}_\tau := \frac{1}{N}\sum_{i=1}^N y_{\tau,i}\boldsymbol{x}_{\tau,i}$, and $(\boldsymbol{x}_\tau, y_\tau) := (\boldsymbol{x}_{\tau,N+1}, y_{\tau,N+1})$. For notational simplicity let us denote $\hat{\boldsymbol{\mu}} := \frac{1}{M}\sum_{i=1}^M y_i\boldsymbol{x}_i$, and let us drop the $M+1$ subscript such that $(\boldsymbol{x}_{M+1}, y_{M+1}) = (\boldsymbol{x}, y)$. Then the test error is given by $\mathbb{P}(\mathrm{sign}(\hat{y}(\boldsymbol{E}; \boldsymbol{W})) \neq y) = \mathbb{P}(\hat{\boldsymbol{\mu}}\boldsymbol{W}y\boldsymbol{x} < 0)$. Using the identity $\sum_{i=1}^M y_i\boldsymbol{x}_i = \boldsymbol{\mu} + \sum_{i=1}^M y_i\boldsymbol{z}_i$ and by properties of the Gausian, we get: $\hat{\boldsymbol{\mu}} \overset{\mathrm{d}}{=} \boldsymbol{\mu} + M^{-1/2}\boldsymbol{z}$ and $y\boldsymbol{x} \overset{\mathrm{d}}{=} \boldsymbol{\mu} + \boldsymbol{z}'$, where $\boldsymbol{z}, \boldsymbol{z}' \overset{\mathrm{i.i.d.}}{\sim} \mathsf{N}(\boldsymbol{0}, \boldsymbol{I}_d)$. Thus, using the transformer prediction rule (Eq. 3):

$$\begin{aligned}
\mathbb{P}(\hat{y}(\boldsymbol{E}; \boldsymbol{W}) \neq y_{M+1}) &= \mathbb{P}\Big( \big(\boldsymbol{\mu} + M^{-1/2}\boldsymbol{z}\big)^\top \boldsymbol{W}(\boldsymbol{\mu} + \boldsymbol{z}') < 0 \Big) \\
&= \mathbb{P}\Big(\boldsymbol{\mu}^\top \boldsymbol{W}\boldsymbol{\mu} < -\boldsymbol{\mu}^\top \boldsymbol{W}\boldsymbol{z}' - M^{-1/2}\boldsymbol{z}^\top\boldsymbol{W}\boldsymbol{\mu} - M^{-1/2}\boldsymbol{z}^\top\boldsymbol{W}\boldsymbol{z}'\Big) \\
&\leq \mathbb{P}\Big(\boldsymbol{\mu}^\top\boldsymbol{W}\boldsymbol{\mu} < \big|\boldsymbol{\mu}^\top\boldsymbol{W}\boldsymbol{z}'\big| + M^{-1/2}\big|\boldsymbol{z}^\top\boldsymbol{W}\boldsymbol{\mu}\big| + \big|M^{-1/2}\boldsymbol{z}^\top\boldsymbol{W}\boldsymbol{z}'\big|\Big)
\end{aligned} \tag{6}$$

Recall that $\boldsymbol{\mu} = \boldsymbol{P}\boldsymbol{\mu}'$, where $\boldsymbol{P} \in \mathbb{R}^{d\times k}$ is semi-orthogonal matrix and $\boldsymbol{\mu}' \overset{\mathrm{i.i.d.}}{\sim} \mathsf{Unif}(R \cdot \mathbb{S}^{k-1})$. Then, we use concentration inequalities of quadratic forms to show that with high probability

$$\boldsymbol{\mu}^\top\boldsymbol{W}\boldsymbol{\mu} \geq \frac{\tilde{R}^2}{k}\,\mathrm{tr}(\boldsymbol{P}^\top\boldsymbol{W}\boldsymbol{P}) - \tilde{O}\left(\frac{\tilde{R}\,\|\boldsymbol{W}\|_F^2}{k}\right), \ \big|\boldsymbol{\mu}^\top\boldsymbol{W}\boldsymbol{z}'\big| \leq \tilde{O}\left(\frac{\tilde{R}\,\|\boldsymbol{W}\|_F}{\sqrt{k}}\right), \ \ \big|\boldsymbol{z}^\top\boldsymbol{W}\boldsymbol{z}'\big| \leq \tilde{O}\left(\|\boldsymbol{W}\|_F\right) \tag{7}$$

Then, our goal becomes: $(i)$ Establish a lower bound on $\mathrm{tr}(\boldsymbol{P}^\top\boldsymbol{W}\boldsymbol{P})$, ensuring that $\boldsymbol{\mu}^\top\boldsymbol{W}\boldsymbol{\mu}$ is large and positive. $(ii)$ Establish an upper bound on $\|\boldsymbol{W}\|_F$. Assuming the number of tasks $B$ is large enough, we can show that $\mathrm{tr}(\boldsymbol{P}^\top\boldsymbol{W}\boldsymbol{P}) = \Omega(k/R^2)$ and $\|\boldsymbol{W}\|_F = O(\sqrt{k}/R^2)$. Substituting these bounds into Eq. 7, and then plugging the result into Eq. 6, yields the desired conclusion.

**Lower bound on** $\mathrm{tr}(\boldsymbol{P}^\top\boldsymbol{W}\boldsymbol{P})$. Using the KKT conditions for the max-margin optimization problem and the fact that $\nabla\hat{y}(\boldsymbol{E}_\tau; \boldsymbol{W}) = \hat{\boldsymbol{\mu}}_\tau\boldsymbol{x}_\tau^\top$, we obtain that there exist $\lambda_1, \ldots, \lambda_B \geq 0$ such that

$$\boldsymbol{W} = \sum_{\tau=1}^B \lambda_\tau y_\tau \hat{\boldsymbol{\mu}}_\tau\boldsymbol{x}_\tau^\top, \tag{8}$$

We first show that $\mathrm{tr}(\boldsymbol{P}^\top\boldsymbol{W}\boldsymbol{P}) \gtrsim R^2 \cdot \sum_{\tau=1}^B \lambda_\tau$, which means it suffices to lower bound $\sum_{\tau=1}^B \lambda_\tau$. Then, by substituting the expression of $\boldsymbol{W}$ (Eq. 8) into the margin constraints (Eq. 5), we obtain that for any $\tau \in [B]$ :

$$1 \leq \hat{\boldsymbol{\mu}}_\tau^\top \left(\sum_{q=1}^B \lambda_q y_q \hat{\boldsymbol{\mu}}_q\boldsymbol{x}_q^\top\right) y_\tau\boldsymbol{x}_\tau = \lambda_\tau\|\hat{\boldsymbol{\mu}}_\tau\|^2\|\boldsymbol{x}_\tau\|^2 + \sum_{q:\, q\neq\tau} \lambda_q\langle\hat{\boldsymbol{\mu}}_\tau, \hat{\boldsymbol{\mu}}_q\rangle\langle y_q\boldsymbol{x}_q, \boldsymbol{x}_\tau y_\tau\rangle.$$

Averaging over $\tau$ and rearranging gives:

$$\begin{aligned}
1 &\leq \frac{1}{B}\sum_{\tau=1}^B \lambda_\tau\|\hat{\boldsymbol{\mu}}_\tau\|^2\|\boldsymbol{x}_\tau\|^2 + \frac{1}{B}\sum_{q=1}^B \lambda_q \sum_{\tau:\tau\neq q}\langle\hat{\boldsymbol{\mu}}_\tau, \hat{\boldsymbol{\mu}}_q\rangle\langle y_q\boldsymbol{x}_q, \boldsymbol{x}_\tau y_\tau\rangle \\
&\leq \sum_{\tau=1}^B \lambda_\tau\frac{\|\hat{\boldsymbol{\mu}}_\tau\|^2\|\boldsymbol{x}_\tau\|^2}{B} + \sum_{q=1}^B \lambda_q \cdot \left|\frac{1}{B}\sum_{\tau:\tau\neq q}\langle\hat{\boldsymbol{\mu}}_\tau, \hat{\boldsymbol{\mu}}_q\rangle\langle y_q\boldsymbol{x}_q, \boldsymbol{x}_\tau y_\tau\rangle\right|.
\end{aligned}$$

To derive a lower bound on $\sum_{\tau=1}^B \lambda_\tau$, we aim first to upper bound $\|\hat{\boldsymbol{\mu}}_\tau\|^2\|\boldsymbol{x}_\tau\|^2/B$, and second upper bound the cross-term average $\left|\frac{1}{B}\sum_{\tau:\tau\neq q}\langle\hat{\boldsymbol{\mu}}_\tau, \hat{\boldsymbol{\mu}}_q\rangle\langle y_q\boldsymbol{x}_q, \boldsymbol{x}_\tau y_\tau\rangle\right|$. While the first is straightforward, the second requires a more delicate argument. Since $\mathbb{E}[\hat{\boldsymbol{\mu}}_\tau] = \mathbb{E}[\boldsymbol{x}_\tau] = \boldsymbol{\mu}_\tau$, the cross-term contains terms like $\langle\boldsymbol{\mu}_\tau, \boldsymbol{\mu}_q\rangle^2$, which can be bounded by $O(R^4/k)$, as well as zero-mean noise terms, whose

average is small when $B$ is large. Together, these imply that both terms are at most $O(R^4/k)$ for large enough $B$, yielding $\sum_{\tau=1}^{B} \lambda_\tau \gtrsim k/R^4$ and thus $\text{tr}(\boldsymbol{P}^\top \boldsymbol{W} \boldsymbol{P}) \gtrsim k/R^2$.

**Upper bound $\|\boldsymbol{W}\|_F$.** We derive an upper bound on $\|\boldsymbol{W}\|_F$ by constructing a matrix $\boldsymbol{U}$ (up to scaling) that satisfies the constraints of the max-margin problem. Since $\boldsymbol{W}$ is the minimum Frobenius norm matrix that separates all training examples, this implies that $\|\boldsymbol{W}\|_F \leq \|\boldsymbol{U}\|_F$. When the signal vector $\boldsymbol{\mu}$ lies in a low-dimensional subspace, a natural candidate for $\boldsymbol{U}$ is the projection matrix $\boldsymbol{P}\boldsymbol{P}^\top$. We show that setting $\boldsymbol{U} := \boldsymbol{P}\boldsymbol{P}^\top$ gives a margin $\hat{\boldsymbol{\mu}}_\tau^\top \boldsymbol{U} y_\tau \boldsymbol{x}_\tau = \tilde{\Theta}(R^2)$, so that $\boldsymbol{U}/R^2$ satisfies the margin constraints. This yields the bound $\|\boldsymbol{W}\|_F \leq \|\boldsymbol{U}/R^2\|_F = \sqrt{k}/R^2$. When the number of tasks $B$ is small (i.e., $B = o(dk/R^2)$), we use an alternative construction: We let $\boldsymbol{U} := \theta \cdot \sum_{q=1}^{B} y_q \hat{\boldsymbol{\mu}}_q \boldsymbol{x}_q^\top$, for a parameter $\theta = \Theta(1/(R^2 d))$. Then we can show that $\|\boldsymbol{W}\|_F \lesssim \sqrt{BR^2/dk} \cdot \sqrt{k}/R^2$. This approach improves the dependence on the number of tasks $B$, compared to Frei and Vardi [11].

**Remark 4.1.** *Interestingly, our analysis indicates that $\boldsymbol{W}_{MM}$ exhibits properties similar to those of the projection matrix $\boldsymbol{P}\boldsymbol{P}^\top$, up to a scaling factor of $R^2$. Specifically, letting $\boldsymbol{U} := \boldsymbol{P}\boldsymbol{P}^\top$, we observe that $\|\boldsymbol{P}^\top \boldsymbol{U} \boldsymbol{P}\|_F = \sqrt{k}$ and $\text{tr}(\boldsymbol{P}^\top \boldsymbol{U} \boldsymbol{P}) = k$. If the matrix $\boldsymbol{W}$ defined by the learning rule in 3 indeed corresponds to this projection, then the transformer effectively carries out the following procedure: it first projects the data onto the ground-truth subspace; next, it performs maximum likelihood estimation (MLE) of the signal $\boldsymbol{\mu}$ by averaging the in-context examples (cf. Example 9.11 in [44]); and finally, it uses this estimate for prediction. Since this is the Bayes classifier under a Gaussian prior (cf. Giraud and Verzelen [13, Appendix B]), this procedure gives an optimal learner with access to the ground-truth representation.*

## 5 Experiments

We complement our theoretical results with an empirical study on metalearning with linear attention. We trained linear attention models (Eq. (3)) on data generated as specified in Section 2.1 using GD with a fixed step size and the logistic loss function. In Figure 1, we compare the in-context sample complexity of linear attention against three baseline algorithms: *(i)* Support Vector Machines (see Section 15 in Shalev-Shwartz and Ben-David [45]); *(ii)* The maximum likelihood estimator (MLE): which estimate $\boldsymbol{\mu}$ under a Gaussian prior by averaging the in-context examples (see Example 9.11 in [44]), and then uses this estimation for prediction; *(iii)* MLE with access to the ground-true matrix $\boldsymbol{P}$, which first projects the data using $\boldsymbol{P}$, and only then applies MLE. We see that the linear transformer outperforms both SVM and MLE, which lack access to $\boldsymbol{P}$, and nearly match the performance of the MLE with projection. Additional experiments and details are provided in the appendix.

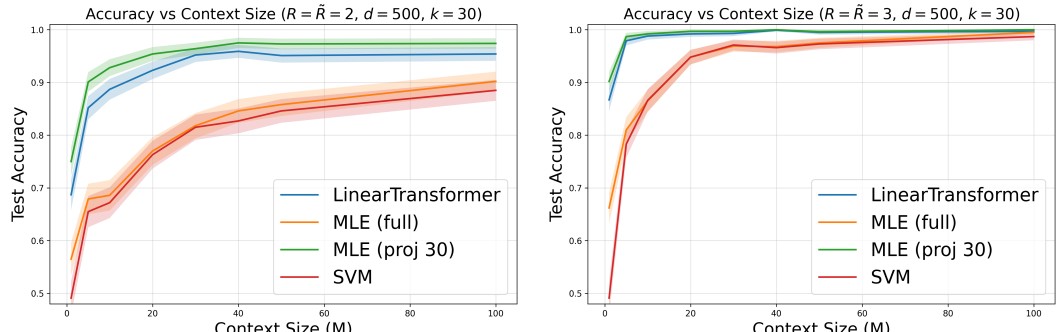

Figure 1: Test accuracy versus the number of in-context examples $M$, where each plot represents a different signal strength $R = \tilde{R}$. We compare the performance of the trained *linear transformer* model against three baselines: full MLE, projected MLE (with access to the true subspace), and SVM. The transformer closely approaches the performance of the projected MLE and outperforms both the full MLE and SVM, which lack access to the subspace. Accuracy improves as the signal strength $R = \tilde{R}$ increases. $d = 500, k = 30, B = 20000$.

# 6 Conclusion and Future Direction

We study the sample complexity of metalearning for the Gaussian mixture framework using a pretrained linear transformer. By analyzing gradient descent, we establish a generalization bound that provably competes with any metalearner and outperforms any algorithm that only has access to the in-context examples. Importantly, our bounds do not depend on the ambient dimension, highlighting the transformer's ability to leverage low-dimensional task structure efficiently.

Our findings underscore the potential of transformers to extract shared representations across diverse but related tasks. This opens several future directions, and encourages extending the metalearning analysis to additional data distributions and transformer architectures, such as deep and multi-head softmax attention.

Moreover, our proof suggests that a trained transformer is closely related to a specific optimal learner with access to the ground-truth representation, namely to a learner that first projects the data onto the ground-truth subspace, and then performs maximum likelihood estimation (see Remark 4.1). However, it remains open whether the transformer can exactly mimic this procedure as the number of training tasks approaches infinity.

Finally, although our results indicate that transformers can implement effective metalearning using only a relatively small number of tasks and examples per task during training, independent of the ambient dimension $d$, an interesting open question is to precisely characterize the minimal sample requirements for successful metalearning. In particular, it remains unclear how many tasks and examples per task are sufficient for training an optimal metalearner, what the exact tradeoff is between the number of tasks and examples per task, and how these requirements may differ between transformer-based architectures and more general metalearning frameworks.

## Acknowledgments and Disclosure of Funding

GV is supported by the Israel Science Foundation (grant No. 2574/25), by a research grant from Mortimer Zuckerman (the Zuckerman STEM Leadership Program), and by research grants from the Center for New Scientists at the Weizmann Institute of Science, and the Shimon and Golde Picker – Weizmann Annual Grant.

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

# Contents

## A  Convex Parameterization of the Linear Transformer

In this section, we provide a more detailed explanation of why our prediction model (Eq. 3) represents a convex parameterization of a linear transformer, a formulation that has been explored in several prior works. Following Eq. 2, The linear transformer with key-query matrix $\boldsymbol{W}^{KQ}$ is defined by

$$f(\boldsymbol{E}; \theta) = \boldsymbol{E} + \boldsymbol{W}^{V} \boldsymbol{E} \cdot \left( \frac{\boldsymbol{E}^{\top} \boldsymbol{W}^{KQ} \boldsymbol{E}}{\rho_{N, d_e}} \right),$$

We use the first $N$ columns of that model to formulate predictions for $\boldsymbol{x}_{N+1}$, whereby the bottom-right corner of the output matrix of $f(\boldsymbol{E}; \theta)$ serves as this prediction.

Writing $\boldsymbol{W}^{\Delta} = \begin{pmatrix} \boldsymbol{W}_{11}^{\Delta} & \boldsymbol{w}_{12}^{\Delta} \\ (\boldsymbol{w}_{21}^{\Delta})^{\top} & w_{22}^{\Delta}, \end{pmatrix}$ for $\Delta \in \{V, KQ\}$, for the linear transformer architecture, this results in the prediction

$$\hat{\boldsymbol{y}}(E; \theta) = \left( (\boldsymbol{w}_{21}^{V})^{\top} \quad w_{22}^{V} \right) \cdot \frac{1}{N} \cdot \boldsymbol{E}\boldsymbol{E}^{\top} \cdot \begin{pmatrix} \boldsymbol{W}_{11}^{KQ} \\ (\boldsymbol{w}_{21}^{KQ})^{\top} \end{pmatrix} \boldsymbol{x}_{N+1}.$$

Due to the product of matrices appearing above, the resulting objective function is non-convex, which makes the analysis of its training dynamics complex. We instead consider a convex parameterization of the linear transformer [10, 9, 11, 12], which results from taking $w_{21}^{KQ} = w_{21}^{V} = 0$ and setting $w_{22}^{V} = 1$. This leads to the following prediction for the label of $\boldsymbol{x}_{N+1}$,

$$\hat{y}(\boldsymbol{E}; \boldsymbol{W}) = \left( \frac{1}{N} \sum_{i=1}^{N} y_i \boldsymbol{x}_i \right)^{\top} \boldsymbol{W} \boldsymbol{x}_{N+1},$$

as defined in Eq. 3.

# B  Proof of Remark 2.2

Let $c, c' > 0$ be the absolute constants from Remark 2.1. Fix a sufficiently large integers $k', d'$ such that $k' \leq d'$, let $\boldsymbol{P}' \in \mathbb{R}^{d' \times k'}$ be a semi-orthogonal matrix, and let $R > 0$ be the signal strength. Consider the dataset $\mathbb{S} = \{(\boldsymbol{x}_1, y_1), \ldots, (\boldsymbol{x}_M, y_M)\} \subseteq \mathbb{R}^{d'} \times \{\pm 1\}$ and the test sample $(\boldsymbol{x}_{M+1}, y_{M+1}) \in \mathbb{R}^{d'} \times \{\pm 1\}$ sampled according to Assumption 2.2 (with $d = d', k = k', \boldsymbol{P} = \boldsymbol{P}'$ and $\tilde{R} = R$). That is, for $\boldsymbol{\mu} \overset{\text{i.i.d.}}{\sim} \boldsymbol{P}' \cdot \text{Unif}(R \cdot \mathbb{S}^{k'-1}), \boldsymbol{z}_i \overset{\text{i.i.d.}}{\sim} \mathsf{N}(\boldsymbol{0}, \boldsymbol{I}_{d'}), y_i \overset{\text{i.i.d.}}{\sim} \text{Unif}(\{\pm 1\})$, we have for any $i \in [M+1]$:

$$\boldsymbol{x}_i \overset{\text{i.i.d.}}{\sim} y_i \boldsymbol{\mu} + \boldsymbol{z}_i. \tag{9}$$

Let $\mathcal{A}^{\boldsymbol{P}'}$ be any algorithm with access to $\boldsymbol{P}'$. Our goal is to show that

$$\Pr_{S, (\boldsymbol{x}_{M+1}, y_{M+1})} [\mathcal{A}^{\boldsymbol{P}'}(S)(\boldsymbol{x}_{M+1}) = y_{M+1}] \geq c \cdot \exp\left( -c' \cdot \min\left\{ R^2, \frac{MR^4}{k} \right\} \right) := \epsilon. \tag{10}$$

Assume by contradiction that this is not the case, i.e,. there exists an algorithm $A^{\boldsymbol{P}}$ whose error is smaller than $\epsilon$. Then we can construct an algorithm $B$ for Gaussian classification with identical spherical covariance and opposite means, as defined in Assumption 2.2 (with $d = k = k'$), that achieves error smaller than $\epsilon$, in contradiction to Remark 2.1. Indeed, given a data set $\mathbb{S}' = \{(\boldsymbol{x}_1', y_1'), \ldots, (\boldsymbol{x}_M', y_M')\} \subseteq \mathbb{R}^{k'} \times \{\pm 1\}$ and a test sample $(\boldsymbol{x}_{M+1}', y_{M+1}') \in \mathbb{R}^{k'} \times \{\pm 1\}$, sampled according to 2.2 (with $k = d = k', \tilde{R} = R$). That is, for $\boldsymbol{\mu}' \overset{\text{i.i.d.}}{\sim} \text{Unif}(R \cdot \mathbb{S}^{k'-1}), \boldsymbol{z}_i \overset{\text{i.i.d.}}{\sim} \mathsf{N}(\boldsymbol{0}, \boldsymbol{I}_k'), y_i' \overset{\text{i.i.d.}}{\sim} \text{Unif}(\{\pm 1\})$, we have for any $i \in [M+1]$:

$$\boldsymbol{x}_i' \overset{\text{i.i.d.}}{\sim} y_i' \boldsymbol{\mu}' + \boldsymbol{z}'.$$

The algorithm $\mathcal{B}$ chooses some semi-orthogonal matrix $\boldsymbol{P}' \in \mathbb{R}^{d' \times k'}$ and sampled independent vectors $\boldsymbol{s}_1, \ldots, \boldsymbol{s}_{M+1} \overset{\text{i.i.d.}}{\sim} \mathsf{N}(\boldsymbol{0}, \boldsymbol{I}_{d'} - \boldsymbol{P}\boldsymbol{P}^{\top})$. Then $\mathcal{B}$ simulates $\mathcal{A}^{\boldsymbol{P}'}$ on the training set $\{(\boldsymbol{P}'\boldsymbol{x}_1' + \boldsymbol{s}_1, y_1'), \ldots, (\boldsymbol{P}'\boldsymbol{x}_M' + \boldsymbol{s}_M, y_M')\}$ and the test sample $(\boldsymbol{P}'\boldsymbol{x}_{M+1} + \boldsymbol{s}_{M+1}, y_{M+1}')$. The key observation is that for any $i \in [M+1]$:

$$\boldsymbol{P}'\boldsymbol{x}_i' + \boldsymbol{s}_i = y_i' \boldsymbol{P}' \boldsymbol{\mu}' + \boldsymbol{P}\boldsymbol{z}_i' + \boldsymbol{s} \overset{\text{d}}{=} \boldsymbol{x}_i,$$

where $\boldsymbol{x}_i$ is from Eq. 9. Indeed, since $\boldsymbol{P}\boldsymbol{z}_i' \sim \mathsf{N}(\boldsymbol{0}, \boldsymbol{P}\boldsymbol{P}^{\top})$, we have $\boldsymbol{P}\boldsymbol{z}_i' + \boldsymbol{s}_i \sim \mathsf{N}(\boldsymbol{0}, \boldsymbol{P}\boldsymbol{P}^{\top} + \boldsymbol{I}_{d'} - \boldsymbol{P}\boldsymbol{P}^{\top}) = \mathsf{N}(\boldsymbol{0}, \boldsymbol{I}_{d'})$. Therefore, we can conclude that

$$\Pr_{S', (\boldsymbol{x}_{M+1}', y_{M+1}')} [\mathcal{B}(S')(\boldsymbol{x}_{M+1}') = y_{M+1}] = \Pr_{S, (\boldsymbol{x}_{M+1}, y_{M+1})} [\mathcal{A}^{\boldsymbol{P}'}(S)(\boldsymbol{x}_{M+1}) = y_{M+1}] < \epsilon,$$

where $\epsilon$ is defined in Eq.10. Contradiction.

# C Proofs for Section 3

## C.1 Notations

First, we introduce useful a notation for the remainder of the proof.

**Assumption C.1.** *For some parameter $c_B > 0$, we have $B = c_B \cdot dk/R^2$.*

The notation $c_B$ from Assumption C.1 specifies how the number of training tasks $B$ scales as a function of the SNR. We will later show that the generalization error on in-context examples depends partly on the quantity $1 \wedge \sqrt{c_B}$, where a larger value implies better generalization. Importantly, we allow $c_B$ to be non-constant; for instance, settings where $B = o_d(d)$ are permitted even for constant $R$.

Table 1: Notation used throughout the paper.

| Symbol | Description |
|--------|-------------|
| $d$ | ambient dimension |
| $k$ | Shared subspace dimension |
| $\boldsymbol{P}$ | Shared low-dimensional subspace & semi-orthogonal matrix $\boldsymbol{P} \in \mathbb{R}^{d \times k}$ |
| $\boldsymbol{\mu}'$ | Cluster mean, $\boldsymbol{\mu}' \in \mathbb{R}^k$ |
| $\boldsymbol{\mu}$ | Isometric embedding of the cluster mean, $\boldsymbol{\mu} = \boldsymbol{P}\boldsymbol{\mu}' \in \mathbb{R}^d$ |
| $\boldsymbol{x}$ | Features, $\boldsymbol{x} \in \mathbb{R}^d$, $\boldsymbol{x}_{\tau,i} = y_{\tau,i}\boldsymbol{\mu}_\tau + \boldsymbol{z}_{\tau,i}$ |
| $y$ | Labels, $y \in \{\pm 1\}$ |
| $\boldsymbol{z}$ | The noise vector, $\boldsymbol{z} \in \mathbb{R}^d$ |
| $\delta$ | Probability of failure |
| $R$ | Norm of cluster means during pre-training |
| $\tilde{R}$ | Norm of cluster means at test time |
| $B$ | Number of pre-training tasks |
| $c_B$ | Quantity such that $B = c_B \cdot dk/R^2$ |
| $N$ | Number of samples per pre-training task |
| $M$ | Number of samples per test-time task |
| $E$ | Data tokanization $E = \begin{pmatrix} x_1 & x_2 & \cdots & x_N & x_{N+1} \\ y_1 & y_2 & \cdots & y_N & 0 \end{pmatrix} \in \mathbb{R}^{(d+1) \times (N+1)}$ |
| $\hat{\boldsymbol{\mu}}$ | Mean predictor: $\frac{1}{M}\sum_{i=1}^{M} y_i \boldsymbol{x}_i$ |
| $\hat{y}(\boldsymbol{E}(x); \boldsymbol{W})$ | Transformer output: $\frac{1}{M}\sum_{i=1}^{M} y_i \boldsymbol{x}_i^T \boldsymbol{W} \boldsymbol{x} = \hat{\mu}^T \boldsymbol{W} \boldsymbol{x}$ |
| $\ell$ | Logistic loss or exponential loss |

## C.2 Properties of the training set

**Lemma C.1.** *Let $\delta \in (0,1)$ be arbitrary. There is an absolute constant $c_0 > 1$ such that with probability at least $1 - \delta$ over the draws of $\{\boldsymbol{\mu}_\tau, (\boldsymbol{x}_\tau, y_\tau), (\boldsymbol{x}_{\tau,i}, y_{\tau,i})_{i=1}^N\}_{\tau=1}^B$, for all $\tau, q \in [B]$ such*

*that $q \neq \tau$ the following hold:*

$$\left| \|\hat{\boldsymbol{\mu}}_\tau\|^2 - R^2 \right| \leq \frac{c_0 R \log(B/\delta)}{\sqrt{N}} + \frac{2d \vee c_0 \log(B/\delta)}{N},$$

$$\left| \|\boldsymbol{x}_\tau\|^2 - d \right| \leq R^2 + c_0 \left( \frac{\log(B/\delta)}{\sqrt{d}} + R \right) \log(B/\delta),$$

$$|\langle \hat{\boldsymbol{\mu}}_q, \hat{\boldsymbol{\mu}}_\tau \rangle| \leq c_0 \left( \frac{R^2}{\sqrt{k}} + \frac{R}{\sqrt{N}} + \frac{\sqrt{d}}{N} \right) \log(B/\delta),$$

$$|\langle \boldsymbol{x}_\tau, \boldsymbol{x}_q \rangle| \leq c_0 \left( \frac{R^2}{\sqrt{k}} + R + \sqrt{d} \right) \log(B/\delta),$$

$$\left| \langle \hat{\boldsymbol{\mu}}_\tau, y_\tau \boldsymbol{x}_\tau \rangle - R^2 \right| \leq c_0 \left( \left[ 1 + \frac{1}{\sqrt{N}} \right] R + \frac{\sqrt{d}}{\sqrt{N}} \right) \log(B/\delta)$$

$$\left| \langle \boldsymbol{P}^\top \hat{\boldsymbol{\mu}}_\tau, \boldsymbol{P}^\top y_\tau \boldsymbol{x}_\tau \rangle - R^2 \right| \leq c_0 \left( \left[ 1 + \frac{1}{\sqrt{N}} \right] R + \frac{\sqrt{k}}{\sqrt{N}} \right) \log(B/\delta)$$

*Proof.* By definition of $\hat{\boldsymbol{\mu}}_\tau$ and properties of the Gaussian distribution, there is $\boldsymbol{z}'_\tau \sim \mathsf{N}(\boldsymbol{0}, \boldsymbol{I}_d)$ such that

$$\hat{\boldsymbol{\mu}}_\tau = \frac{1}{N} \sum_{i=1}^N y_{\tau,i}(y_{\tau,i}\boldsymbol{\mu}_\tau + \boldsymbol{z}_{\tau,i}) = \boldsymbol{\mu}_\tau + \frac{1}{N} \sum_{i=1}^N y_{\tau,i}\boldsymbol{z}_{\tau,i} = \boldsymbol{\mu}_\tau + \frac{1}{\sqrt{N}}\boldsymbol{z}'_\tau.$$

Thus for $\tau \neq q$ there are $\boldsymbol{z}'_\tau, \boldsymbol{z}'_q \overset{\text{i.i.d.}}{\sim} \mathsf{N}(\boldsymbol{0}, \boldsymbol{I}_d)$ such that

$$\langle \hat{\boldsymbol{\mu}}_q, \hat{\boldsymbol{\mu}}_\tau \rangle \overset{\mathrm{d}}{=} \langle \boldsymbol{\mu}_\tau + N^{-1/2}\boldsymbol{z}'_\tau, \boldsymbol{\mu}_q + N^{-1/2}\boldsymbol{z}'_q \rangle$$
$$= \langle \boldsymbol{\mu}_\tau, \boldsymbol{\mu}_q \rangle + N^{-1/2}\langle \boldsymbol{z}'_\tau, \boldsymbol{\mu}_q \rangle + N^{-1/2}\langle \boldsymbol{z}'_q, \boldsymbol{\mu}_\tau \rangle + N^{-1}\langle \boldsymbol{z}'_\tau, \boldsymbol{z}'_q \rangle. \tag{11}$$

We first derive an upper bound for this quantity when $q \neq \tau$. Remember that $\boldsymbol{\mu}_\tau = \boldsymbol{P}\boldsymbol{\mu}'_\tau$ for some semi-orthogonal matrix $\boldsymbol{P}$ and $\boldsymbol{\mu}'_\tau \sim \mathsf{Unif}(\mathbb{S}^{k-1})$.

- $|\langle \boldsymbol{\mu}_\tau, \boldsymbol{\mu}_q \rangle|$ **Analysis.** since $\boldsymbol{\mu}'_q, \boldsymbol{\mu}'_\tau$ are independent and sub-Gaussian random vectors with sub-Gaussian norm at most $cR/\sqrt{k}$ (Remark D.1) for some absolute constant $c > 0$, by Vershynin [46, Lemma 6.2.3] with $\boldsymbol{A} = \boldsymbol{I}_k$, we have for some $c' > 0$ it holds that for any $\beta \in \mathbb{R}$, if $\boldsymbol{g}, \boldsymbol{g}' \overset{\text{i.i.d.}}{\sim} \mathsf{N}(0, I_k)$,

$$\mathbb{E}[\exp(\beta\boldsymbol{\mu}_q^\top \boldsymbol{\mu}_\tau)] = \mathbb{E}[\exp(\beta\boldsymbol{\mu}'^\top_q \boldsymbol{\mu}'_\tau)] \leq \mathbb{E}[\exp(c'R^2 k^{-1}\beta\boldsymbol{g}^\top \boldsymbol{g}')].$$

  By Vershynin [46, Lemma 6.2.2], for some $c_1 > 0$, provided $c'|\beta|R^2/k \leq c_1$, it holds that

$$\mathbb{E}[\exp(c'R^2 k^{-1}\beta\boldsymbol{g}^\top \boldsymbol{g}')] \leq \exp(c_1\beta^2 R^4 k^{-2}\|I_k\|_F^2) = \exp(c_1\beta^2 R^4 k^{-1}).$$

  Since $\boldsymbol{\mu}_q, \boldsymbol{\mu}_\tau$ are mean-zero, by Vershynin [46, Proposition 2.7.1] this implies the quantity $\boldsymbol{\mu}_q^\top \boldsymbol{\mu}_\tau$ is sub-exponential with $\|\boldsymbol{\mu}_q^\top \boldsymbol{\mu}_\tau\|_{\psi_1} \leq c_2 R^2/\sqrt{k}$ for some absolute constant $c_2 > 0$. We therefore have by definition of sub-exponential [46, Proposition 2.7.1, first item] and union bound, that for some absolute constant $c_3 > 0$, w.p. at least $1 - \delta$, for all $\tau \neq q$,

$$|\langle \boldsymbol{\mu}_\tau, \boldsymbol{\mu}_q \rangle| \leq c_3 R^2 k^{-1/2} \log(B/\delta). \tag{12}$$

- $\boldsymbol{\mu}_q^\top \boldsymbol{z}_\tau$ **Analysis.** We have that $\boldsymbol{\mu}_q^\top \boldsymbol{z}_\tau = \boldsymbol{\mu}'_q \boldsymbol{P}^\top \boldsymbol{z}'_q$. Since $\boldsymbol{\mu}'_q$ has sub-Gaussian norm at most $cR/\sqrt{k}$ (Remark D.1) and $\boldsymbol{P}^\top \boldsymbol{z}'_q$ has sub-Gaussian norm at most $c$ (Remark D.2), by using again Lemmas 6.2.2 and 6.2.3 from [46], we have for any $\beta \in \mathbb{R}$, if $\boldsymbol{g}, \boldsymbol{g}' \overset{\text{i.i.d.}}{\sim} \mathsf{N}(0, I_k)$, then

$$\mathbb{E}[\exp(\beta\boldsymbol{\mu}_q^\top \boldsymbol{z}'_\tau)] \leq \mathbb{E}[\exp(c'Rk^{-1/2}\beta\boldsymbol{g}^\top \boldsymbol{g}')],$$

and thus provided $c'Rk^{-1/2}|\beta| \leq c_1$ we have

$$\mathbb{E}[\exp(c'Rk^{-1/2}\beta \boldsymbol{g}^\top \boldsymbol{g}')] \leq \exp(c_1 R^2 k^{-1} \beta^2 \|\boldsymbol{I}_k\|_F^2) = \exp(c_1 R^2 \beta^2).$$

In particular, the quantity $\boldsymbol{\mu}_q^\top \boldsymbol{z}'_\tau$ is sub-exponential with sub-exponential norm $\|\boldsymbol{\mu}_q^\top \boldsymbol{z}'_\tau\|_{\psi_1} \leq c_2 R$, and so for some absolute constant $c_3 > 0$ we have with probability at least $1 - \delta$, for all $q, \tau \in [B]$ with $q \neq \tau$,

$$|\langle \boldsymbol{\mu}_q, \boldsymbol{z}'_\tau \rangle| \leq c_3 R \log(B/\delta). \tag{13}$$

- $\langle \boldsymbol{z}'_q, \boldsymbol{z}'_\tau \rangle$ **Analysis.** For $\langle \boldsymbol{z}'_q, \boldsymbol{z}'_\tau \rangle$ with $\tau \neq q$ we can directly use the MGF of Gaussian chaos [46, Lemma 6.2.2]: $\langle \boldsymbol{z}'_q, \boldsymbol{z}'_\tau \rangle = \boldsymbol{g}^\top \boldsymbol{I}_d \boldsymbol{g}'$ for i.i.d. $\boldsymbol{g}, \boldsymbol{g}' \sim \mathsf{N}(\boldsymbol{0}, \boldsymbol{I}_d)$ so that for $\beta \leq c/\|\boldsymbol{I}_d\|_2$,

$$\mathbb{E}[\exp(\beta \langle \boldsymbol{z}'_q, \boldsymbol{z}'_\tau \rangle)] \leq \exp(c_4 \beta^2 \|\boldsymbol{I}_d\|_F^2) = \exp(c_4 \beta^2 d).$$

In particular, $\left\|\langle \boldsymbol{z}'_q, \boldsymbol{z}'_\tau \rangle\right\|_{\psi_1} \leq c_5 \sqrt{d}$ so that sub-exponential concentration implies that with probability at least $1 - \delta$, for any $q, \tau \in [B]$ with $q \neq \tau$,

$$|\langle \boldsymbol{z}'_\tau, \boldsymbol{z}'_q \rangle| \leq c_6 \sqrt{d} \log(B/\delta). \tag{14}$$

Putting Eq. 12, Eq. 13, and Eq. 14 into Eq. 11 we get for $q \neq \tau$,

$$|\langle \hat{\boldsymbol{\mu}}_q, \hat{\boldsymbol{\mu}}_\tau \rangle| = c_7 \left( \frac{R^2}{\sqrt{k}} + \frac{R}{\sqrt{N}} + \frac{\sqrt{d}}{N} \right) \log(B/\delta). \tag{15}$$

As for $\|\hat{\boldsymbol{\mu}}_\tau\|^2$, from Eq. 11 we have

$$\|\hat{\boldsymbol{\mu}}_\tau\|^2 = \|\boldsymbol{\mu}_\tau\|^2 + 2N^{-1/2}\langle \boldsymbol{z}'_\tau, \boldsymbol{\mu}_\tau \rangle + N^{-1}\|\boldsymbol{z}'_\tau\|^2 \tag{16}$$

From here, the same argument used to bound Eq. 13 holds since that bound only relied upon the fact that $\boldsymbol{\mu}_q$ and $\boldsymbol{z}'_\tau$ are independent, while $\boldsymbol{\mu}_\tau$ and $\boldsymbol{z}'_\tau$ are independent as well. In particular, with probability at least $1 - \delta$, for all $\tau \in [B]$,

$$|\langle \boldsymbol{\mu}_\tau, \boldsymbol{z}'_\tau \rangle| \leq c_3 R \log(B/\delta). \tag{17}$$

Each coordinate of $\boldsymbol{z}'_\tau$ has sub-exponential norm at most some constant $c$ and $\mathbb{E}[\|\boldsymbol{z}'_\tau\|^2] = d$. Therefore by by Bernstein's inequality [46, Thm. 2.8.1], we have for some constant $c_9 > 0$, with probability at least $1 - \delta$, for any $\tau \in [B]$,

$$\left|\|\boldsymbol{z}'_\tau\|^2 - d\right| \leq c_9 \sqrt{\frac{\log(B/\delta)}{d}}. \tag{18}$$

Putting Eq. 18 and Eq. 17 into Eq. 16 and using that $\|\boldsymbol{\mu}_\tau\|^2 = R^2$, we get with probability at least $1 - 2\delta$,

$$\left|\|\hat{\boldsymbol{\mu}}_\tau\|^2 - R^2\right| \leq \frac{c_5 R \log(2B/\delta)}{\sqrt{N}} + \frac{2d \vee c_9 \log(B/\delta)}{N}.$$

As for $\|\boldsymbol{x}_\tau\|^2$, by definition,

$$\|\boldsymbol{x}_\tau\|^2 = \|\boldsymbol{\mu}_\tau\|^2 + 2\langle \boldsymbol{\mu}_\tau, \boldsymbol{z}_\tau \rangle + \|\boldsymbol{z}_\tau\|^2 = R^2 + 2\langle \boldsymbol{\mu}_\tau, \boldsymbol{z}_\tau \rangle + \|\boldsymbol{z}_\tau\|^2.$$

Since $\boldsymbol{z}_\tau \sim \mathsf{N}(\boldsymbol{0}, \boldsymbol{I}_d)$ has the same distribution as $\boldsymbol{z}'_\tau$, the same analysis used to prove Eq. 17 and Eq. 18 yields that with probability at least $1 - 2\delta$, for all $\tau \in [B]$,

$$|\langle \boldsymbol{\mu}_\tau, \boldsymbol{z}_\tau \rangle| \leq c_3 R \log(B/\delta),$$

$$\|\boldsymbol{z}_\tau\|^2 \leq d + \frac{c_9 \log(B/\delta)}{\sqrt{d}}.$$

Substituting these into the preceding display we have that

$$\left|\|\boldsymbol{x}_\tau\|^2 - d\right| \leq R^2 + \frac{c_9 \log(B/\delta)}{\sqrt{d}} + c_3 R \log(B/\delta)$$

Thus provided $d$ is sufficiently large, then we also have $\|\boldsymbol{x}_\tau\|^2 \simeq d$.

Next we bound $|\langle \boldsymbol{x}_\tau, \boldsymbol{x}_q \rangle|$: There are $\boldsymbol{z}'_\tau, \boldsymbol{z}'_q \overset{\text{i.i.d.}}{\sim} \mathsf{N}(\boldsymbol{0}, \boldsymbol{I}_d)$ such that

$$\langle y_\tau \boldsymbol{x}_\tau, y_q \boldsymbol{x}_q \rangle \overset{\text{d}}{=} \langle \boldsymbol{\mu}_\tau + \boldsymbol{z}'_\tau, \boldsymbol{\mu}_q + \boldsymbol{z}'_q \rangle.$$

It is clear that the same exact analysis we used to analyze Eq. 11 leads to the claim that with probability at least $1 - \delta$, for all $q \neq \tau$:

$$|\langle \boldsymbol{x}_q, \boldsymbol{x}_\tau \rangle| \leq c_9 \left( \frac{R^2}{\sqrt{k}} + R + \sqrt{d} \right) \log(B/\delta). \tag{19}$$

Finally, we consider $y_\tau \hat{\boldsymbol{\mu}}_\tau^\top \boldsymbol{x}_\tau$. Just as in the previous analyses, there are $\boldsymbol{z}_\tau, \boldsymbol{z}'_\tau \sim \mathsf{N}(\boldsymbol{0}, \boldsymbol{I}_d)$ such that

$$\langle \hat{\boldsymbol{\mu}}_\tau, y_\tau \boldsymbol{x}_\tau \rangle \overset{\text{d}}{=} \langle \boldsymbol{\mu}_\tau + N^{-1/2} \boldsymbol{z}_\tau, \boldsymbol{\mu}_\tau + \boldsymbol{z}'_\tau \rangle$$
$$= \|\boldsymbol{\mu}_\tau\|^2 + N^{-1/2} \langle \boldsymbol{z}_\tau, \boldsymbol{\mu}_\tau \rangle + \langle \boldsymbol{z}'_\tau, \boldsymbol{\mu}_\tau \rangle + N^{-1/2} \langle \boldsymbol{z}_\tau, \boldsymbol{z}'_\tau \rangle.$$

Again using an analysis similar to that used for Eq. 11 yields that with probability at least $1 - \delta$, for all $\tau \in [B]$,

$$\left| \langle \hat{\boldsymbol{\mu}}_\tau, y_\tau \boldsymbol{x}_\tau \rangle - R^2 \right| \leq c_{10} \left( \left[ 1 + \frac{1}{\sqrt{N}} \right] R + \frac{\sqrt{d}}{\sqrt{N}} \right) \log(2B/\delta). \tag{20}$$

Moreover, note that $\langle \boldsymbol{P}^\top \hat{\boldsymbol{\mu}}_\tau, \boldsymbol{P}^\top y_\tau \boldsymbol{x}_\tau \rangle \overset{\text{d}}{=} \langle \boldsymbol{\mu}'_\tau + \boldsymbol{z}_\tau/\sqrt{n}, \boldsymbol{\mu}'_\tau + \boldsymbol{z}'_\tau \rangle$ for $\boldsymbol{z}_\tau, \boldsymbol{z}'_\tau \overset{\text{i.i.d.}}{\sim} N(\boldsymbol{0}, \boldsymbol{I}_k)$ (see Remark D.2). Then we can use the same argument as Eq. 20 with $k = d$ and $\boldsymbol{P} = \boldsymbol{I}_k$ to conclude that with probability at least $1 - \delta$ we have:

$$\left| \langle \boldsymbol{P}^\top \hat{\boldsymbol{\mu}}_\tau, \boldsymbol{P}^\top y_\tau \boldsymbol{x}_\tau \rangle - R^2 \right| \leq c_{10} \left( \left[ 1 + \frac{1}{\sqrt{N}} \right] R + \frac{\sqrt{k}}{\sqrt{N}} \right) \log(2B/\delta). \tag{21}$$

Taking a union bound over each of the events shows that all of the desired claims of Lemma C.1 hold with probability at least $1 - 10\delta$. It is also easy to verify that the Lemma holds with probability at least $1 - \delta$, for a different choice of constant $c_0$.

$\square$

**Lemma C.2.** *Let $\delta \in (0, 1)$ be arbitrary. There is an absolute constant $c_0 > 1$ such that with probability at least $1 - \delta$ over the draws of $\{\boldsymbol{\mu}_\tau, (\boldsymbol{x}_\tau, y_\tau), (\boldsymbol{x}_{\tau,i}, y_{\tau,i})_{i=1}^N\}_{\tau=1}^B$, then we have*

$$\operatorname{tr}(\boldsymbol{P}^\top \boldsymbol{W}_{MM} \boldsymbol{P}) \geq \left( R^2 - c_0 \left( \left[ 1 + \frac{1}{\sqrt{N}} \right] R + \frac{\sqrt{d}}{\sqrt{N}} \right) \log(B/\delta) \right) \sum_{q=1}^B \lambda_q$$

*Proof.* By Remark D.2

$$\boldsymbol{P}^\top \boldsymbol{W}_{\text{MM}} \boldsymbol{P} = \sum_{q=1}^B \lambda_q \boldsymbol{P}^\top \hat{\boldsymbol{\mu}}_q y_q \boldsymbol{x}_q^\top \boldsymbol{P} = \sum_{q=1}^B \lambda_q (\boldsymbol{\mu}'_q + \boldsymbol{z}_q/\sqrt{N})(\boldsymbol{\mu}_q'^\top + \boldsymbol{z}_q'^\top),$$

where $\boldsymbol{z}_q, \boldsymbol{z}'_q \overset{\text{i.i.d.}}{\sim} N(\boldsymbol{0}, \boldsymbol{I}_k)$. Next, we lower bound $\operatorname{tr}(\boldsymbol{P}^\top \boldsymbol{W}_{\text{MM}} \boldsymbol{P})$ as a function of $\sum_{q=1}^B \lambda_q$:

$$\operatorname{tr}(\boldsymbol{P}^\top \boldsymbol{W}_{\text{MM}} \boldsymbol{P}) = \sum_{q=1}^B \lambda_q \operatorname{tr} \left( \left( \boldsymbol{\mu}'_q + \boldsymbol{z}_q/\sqrt{N} \right) \left( \boldsymbol{\mu}_q'^\top + \boldsymbol{z}_q'^\top \right) \right). \tag{22}$$

Observe that

$$\operatorname{tr} \left( \left( \boldsymbol{\mu}'_q + \boldsymbol{z}_q/\sqrt{N} \right) \left( \boldsymbol{\mu}_q'^\top + \boldsymbol{z}_q'^\top \right) \right) = \operatorname{tr} \left( \left( \boldsymbol{\mu}_q'^\top + \boldsymbol{z}_q'^\top \right) \left( \boldsymbol{\mu}'_q + \boldsymbol{z}_q/\sqrt{N} \right) \right)$$
$$\geq R^2 - c_0 \left( \left[ 1 + \frac{1}{\sqrt{N}} \right] R + \frac{\sqrt{d}}{\sqrt{N}} \right) \log(B/\delta),$$

where the last inequality follows from Lemma C.1 (last item), and since for $k = d$ and $\boldsymbol{P} = \boldsymbol{I}_k$, we have that $\langle \hat{\boldsymbol{\mu}}_q, y_q \boldsymbol{x}_q \rangle \overset{\text{d}}{=} \left( \boldsymbol{\mu}_q'^\top + \boldsymbol{z}_q'^\top \right) \left( \boldsymbol{\mu}'_q + \boldsymbol{z}_q/\sqrt{N} \right)$.

Substituting the displayed Eq. into Eq. 22 yields the desired result. $\square$

The events of Lemmas C.1 and C.2 hold with probability at least $1 - \delta$, independently of Assumption A.[2] Combining Lemmas C.1, C.2 and Assumption A we can conclude that:

**Lemma C.3.** *Suppose that Assumption A holds for sufficiently large $C$. Then there exists some constant $c_0$ such that with probability at least $1 - 5\delta$:*

$$\left| \|\hat{\boldsymbol{\mu}}_\tau\|^2 - R^2 \right| \leq R^2/4, \tag{23}$$

$$\left| \|\boldsymbol{x}_\tau\|^2 - d \right| \leq d/4, \tag{24}$$

$$|\langle \hat{\boldsymbol{\mu}}_q, \hat{\boldsymbol{\mu}}_\tau \rangle| \leq c_0 \left( \frac{R^2}{\sqrt{k}} \right) \log(B/\delta), \tag{25}$$

$$|\langle \boldsymbol{x}_\tau, \boldsymbol{x}_q \rangle| \leq c_0 \left( \frac{R^2}{\sqrt{k}} \vee \sqrt{d} \right) \log(B/\delta), \tag{26}$$

$$\left| \langle \hat{\boldsymbol{\mu}}_\tau, y_\tau \boldsymbol{x}_\tau \rangle - R^2 \right| \leq R^2/4 \tag{27}$$

$$\operatorname{tr}(\boldsymbol{P}^\top \boldsymbol{W}_{MM} \boldsymbol{P}) \geq \frac{R^2}{2} \sum_{q=1}^{B} \lambda_q \tag{28}$$

$$\left| \langle \boldsymbol{P}^\top \hat{\boldsymbol{\mu}}_\tau, \boldsymbol{P}^\top y_\tau \boldsymbol{x}_\tau \rangle - R^2 \right| \leq \frac{R^2}{4} \tag{29}$$

*Moreover, recall that $B = c_B \cdot dk/R^2$ (Assumption C.1). Then for any $\tau \in [B]$ we have*

$$\frac{\|\hat{\boldsymbol{\mu}}_\tau\|^2 \|\boldsymbol{x}_\tau\|^2}{B} \leq \frac{2R^4}{c_B k} \tag{30}$$

$$\left| \frac{1}{B} \sum_{q \in [B]: q \neq \tau} \langle \hat{\boldsymbol{\mu}}_\tau, \hat{\boldsymbol{\mu}}_q \rangle \langle y_q \boldsymbol{x}_q, \boldsymbol{x}_\tau y_\tau \rangle \right| \leq c \left( \frac{1}{R\sqrt{c_B}} \vee 1 \right) \frac{R^4}{k} \log^2(B/\delta) \leq c \left( \frac{1}{c_B} \vee 1 \right) \frac{R^4}{k} \log^2(B/\delta), \tag{31}$$

*for some constant $c > 0$. We can also conclude that if $B \leq dk$, i.e. $\sqrt{c_B} \leq 1/R$, then we have that*

$$\left| \sum_{q \in [B]: q \neq \tau} \langle \hat{\boldsymbol{\mu}}_\tau, \hat{\boldsymbol{\mu}}_q \rangle \langle y_q \boldsymbol{x}_q, \boldsymbol{x}_\tau y_\tau \rangle \right| \leq c \left( \frac{B}{R\sqrt{c_B}} \right) \frac{R^4}{k} \log^2(B/\delta)$$

$$= c \cdot \left( \frac{R^2 \sqrt{Bd}}{\sqrt{k}} \right) \log^2(B/\delta) \tag{32}$$

*Proof.* The first part (Eqs. 23-29) occurs with probability at least $1 - 2\delta$, and follows directly by substituting Assumption A into Lemmas C.1 and C.2, and by applying the union bound. Regarding the last part, observe that

$$\frac{\|\hat{\boldsymbol{\mu}}_\tau\|^2 \|\boldsymbol{x}_\tau\|^2}{B} \leq \frac{2R^2 d}{B} = 2\frac{R^4}{c_B k}, \tag{33}$$

where the first inequality holds by the first part of the lemma, i.e. the upper bounds on $\|\hat{\boldsymbol{\mu}}_\tau\|^2, \|\boldsymbol{x}_\tau\|^2$. The last equality holds for by definition of $B$. This proves Eq. 30. Moreover, observe that

$$\left| \frac{1}{B} \sum_{q \in [B]: q \neq \tau} \langle \hat{\boldsymbol{\mu}}_\tau, \hat{\boldsymbol{\mu}}_q \rangle \langle y_q \boldsymbol{x}_q, \boldsymbol{x}_\tau y_\tau \rangle \right| \leq \left| \frac{1}{B} \sum_{q \in [B]: q \neq \tau} \langle \hat{\boldsymbol{\mu}}_\tau, \hat{\boldsymbol{\mu}}_q \rangle \langle \boldsymbol{\mu}_q, \boldsymbol{\mu}_\tau \rangle \right|$$

$$+ \left| \frac{1}{B} \sum_{q \in [B]: q \neq \tau} \langle \hat{\boldsymbol{\mu}}_\tau, \hat{\boldsymbol{\mu}}_q \rangle \langle \boldsymbol{\mu}_q, \boldsymbol{z}_\tau \rangle \right| + \left| \frac{1}{B} \sum_{q \in [B]: q \neq \tau} \langle \hat{\boldsymbol{\mu}}_\tau, \hat{\boldsymbol{\mu}}_q \rangle \langle \boldsymbol{z}_q, \boldsymbol{\mu}_\tau \rangle \right|$$

$$+ \left| \frac{1}{B} \sum_{q \in [B]: q \neq \tau} \langle \hat{\boldsymbol{\mu}}_\tau, \hat{\boldsymbol{\mu}}_q \rangle \langle \boldsymbol{z}_q, \boldsymbol{z}_\tau \rangle \right|. \tag{34}$$

---

[2]Note, however, that the quantities appearing on the right-hand sides of each inequality in the lemma are only small when these assumptions hold; this is the reason for these assumptions.

Next, we analyze each of the above terms seperately. Regarding the first term,

$$\left| \frac{1}{B} \sum_{q \in [B]: q \neq \tau} \langle \hat{\boldsymbol{\mu}}_\tau, \hat{\boldsymbol{\mu}}_q \rangle \langle \boldsymbol{\mu}_q, \boldsymbol{\mu}_\tau \rangle \right| \leq \max_{\tau, q} |\langle \hat{\boldsymbol{\mu}}_\tau, \hat{\boldsymbol{\mu}}_q \rangle| \, |\langle \boldsymbol{\mu}_q, \boldsymbol{\mu}_\tau \rangle| \leq c_0 \frac{R^4}{k} \log^2(B/\delta). \quad (35)$$

where the last inequality holds by the first part (third item) of this Lemma and Eq. 12. Regarding the second term of Eq. 34, let's fix $\hat{\boldsymbol{\mu}}_\tau, \hat{\boldsymbol{\mu}}_q$ and $\boldsymbol{\mu}_q$ and observe that $\langle \hat{\boldsymbol{\mu}}_\tau, \hat{\boldsymbol{\mu}}_q \rangle \langle \boldsymbol{\mu}_q, \boldsymbol{z}_1 \rangle, \dots, \langle \hat{\boldsymbol{\mu}}_\tau, \hat{\boldsymbol{\mu}}_q \rangle \langle \boldsymbol{\mu}_q, \boldsymbol{z}_B \rangle$ are independent random variables. We emphasize that we currently treat only $\boldsymbol{z}_1, \dots, \boldsymbol{z}_B$ as random variables, while the other terms are fixed. In other words, the following analysis holds for any realization of $\hat{\boldsymbol{\mu}}_\tau, \hat{\boldsymbol{\mu}}_q$ and $\boldsymbol{\mu}_q$, conditioning on the events of the first part (Eqs. 23-29). By General Hoeffding's inequality (Thm. 2.6.3 from Vershynin [46]), we have

$$\mathbb{P}\left( \left| \frac{1}{B} \sum_{q \in [B]: q \neq \tau} \langle \hat{\boldsymbol{\mu}}_\tau, \hat{\boldsymbol{\mu}}_q \rangle \langle \boldsymbol{\mu}_q, \boldsymbol{z}_\tau \rangle \right| \geq t \right) \leq 2 \exp\left( -\frac{c t^2 B^2}{K^2 \sum_{q: q \neq \tau} \langle \hat{\boldsymbol{\mu}}_\tau, \hat{\boldsymbol{\mu}}_q \rangle^2} \right),$$

where $c$ is some constant and $K := \max_{q: q \neq \tau} \|\langle \boldsymbol{\mu}_q, \boldsymbol{z}_\tau \rangle\|_{\psi_2}$. By properties of the Gaussian distribution we have that $\langle \boldsymbol{\mu}_q, \boldsymbol{z}_\tau \rangle \sim \mathsf{N}(\mathbf{0}, \|\boldsymbol{\mu}_q\|^2)$, which means that $\|\langle \boldsymbol{\mu}_q, \boldsymbol{z}_\tau \rangle\|_{\psi_2} = c \|\boldsymbol{\mu}_q\| = cR$, for some constant $c$. Moreover, by the first part of this lemma (item 3) we have that $\sum_{q: q \neq \tau} \langle \hat{\boldsymbol{\mu}}_\tau, \hat{\boldsymbol{\mu}}_q \rangle^2 \leq c_0 B R^4 \log^2(B/\delta)/k$. By choosing $t = \sqrt{R^6 \log(B/\delta) \log^2(2/\delta)/cBk}$, we obtain that with probability at least $1 - \delta$, for any $\tau \in [B]$,

$$\left| \frac{1}{B} \sum_{q \in [B]: q \neq \tau} \langle \hat{\boldsymbol{\mu}}_\tau, \hat{\boldsymbol{\mu}}_q \rangle \langle \boldsymbol{\mu}_q, \boldsymbol{z}_\tau \rangle \right| \leq \sqrt{\frac{R^6 \log^2(B/\delta) \log(2/\delta)}{cBk}} \leq \frac{1}{c \cdot \sqrt{c_B d}} \cdot \frac{R^4}{k} \cdot \log^2(B/\delta), \quad (36)$$

where the last inequality holds by definition of $B$. The same bound also holds for the third term in Eq. 34. Regarding the last term of Eq. 34, we can use again the General Hoeffding's inequality to obtain,

$$\mathbb{P}\left( \left| \frac{1}{B} \sum_{q \in [B]: q \neq \tau} \langle \hat{\boldsymbol{\mu}}_\tau, \hat{\boldsymbol{\mu}}_q \rangle \langle \boldsymbol{z}_q, \boldsymbol{z}_\tau \rangle \right| \geq t \right) \leq 2 \exp\left( -\frac{c t^2 B^2}{K^2 \sum_{q: q \neq \tau} \langle \hat{\boldsymbol{\mu}}_\tau, \hat{\boldsymbol{\mu}}_q \rangle^2} \right),$$

where now $K := \max_{q: q \neq \tau} \|\langle \boldsymbol{z}_q, \boldsymbol{z}_\tau \rangle\|_{\psi_2}$. Now we have that $\langle \boldsymbol{z}_q, \boldsymbol{z}_\tau \rangle \sim \mathsf{N}(\mathbf{0}, \|\boldsymbol{z}_\tau\|^2)$, which means that $\|\langle \boldsymbol{z}_q, \boldsymbol{z}_\tau \rangle\|_{\psi_2} = \|\boldsymbol{z}_\tau\| \leq 2c\sqrt{d}$. By choosing $t = \sqrt{dR^4 \log^2(B/\delta) \log(2/\delta)/cBk}$, we obtain that with probability at least $1 - \delta$,

$$\left| \frac{1}{B} \sum_{q \in [B]: q \neq \tau} \langle \hat{\boldsymbol{\mu}}_\tau, \hat{\boldsymbol{\mu}}_q \rangle \langle \boldsymbol{z}_q, \boldsymbol{z}_\tau \rangle \right| \leq \sqrt{\frac{dR^4 \log^2(B/\delta) \log(2/\delta)}{cBk}} \leq \frac{1}{cR \cdot \sqrt{c_B}} \cdot \frac{R^4}{k} \log^2(B/\delta), \quad (37)$$

where the last inequality holds by definition of $B$. Substituting Eqs. 35, 36 and 37 into Eq. 34, and observe that $1 \vee 1/c_B \geq 1/R\sqrt{c_B} \vee 1/\sqrt{dc_B}$, for any $R, d = \Omega(1)$ (Assumptions (A2) and (A3)), Eq. 31 follows. We note that by union bound, all the events of this lemma occur with probability at least $1 - 5\delta$. $\qquad\square$

Our results will require this event to hold, so we introduce the following to allow us to refer to it in later lemmas:

**Definition C.1.** *Let us say that a good run occurs if the events of Lemma C.3 hold. By that lemma, this happens with probability at least $1 - 5\delta$ over the draws of the training sets.*

## C.3 Analysis of $W_{\mathrm{MM}}$

Next, we introduce the following lemma regarding the max-margin solution $\boldsymbol{W} := \boldsymbol{W}_{\mathrm{MM}}$, as defined in Eq. 4:

**Lemma C.4.** *On a good run and for $C > 1$ sufficiently large under Assumption A , the max-margin solution $\boldsymbol{W}$ of Problem 4,*

$$\boldsymbol{W} = \sum_{\tau=1}^{B} \lambda_\tau y_\tau \hat{\boldsymbol{\mu}}_\tau \boldsymbol{x}_\tau^\top,$$

*is such that the $\lambda_\tau \geq 0$ satisfy the following:*

$$\sum_{\tau=1}^{B} \lambda_\tau \geq c \cdot \frac{(c_B \wedge 1)}{\log^2(B/\delta)} \cdot \frac{k}{R^4},$$

*where $c_0 > 1$ is some constant. Further, we have the inequalities*

$$\|\boldsymbol{P}^\top \boldsymbol{W}\boldsymbol{P}\|_F \wedge \|\boldsymbol{W}\|_F \leq (1 \wedge \sqrt{c_B}) \cdot \frac{\sqrt{k}}{R^2},$$

*and*

$$\mathrm{tr}(\boldsymbol{P}^\top \boldsymbol{W}\boldsymbol{P}) \geq c \cdot \frac{(c_B \wedge 1)}{\log^2(B/\delta)} \cdot \frac{k}{R^2},$$

*for some constant $c$. We recall that $B = c_B \cdot dk/R^2$ (Assumption C.1).*

*Proof.* In this part, $c$ and $c_0$ represent some constants that can change from line to line.

- $\mathrm{tr}(\boldsymbol{P}^\top \boldsymbol{W}\boldsymbol{P})$ **lower bound.** By the feasibility conditions of the max-margin problem, we have for any $\tau \in [B]$,

$$1 \leq y_\tau \hat{\boldsymbol{\mu}}_\tau^\top \boldsymbol{W}\boldsymbol{x}_\tau$$

$$= \hat{\boldsymbol{\mu}}_\tau^\top \left( \sum_{q=1}^{B} \lambda_q y_q \hat{\boldsymbol{\mu}}_q \boldsymbol{x}_q^\top \right) y_\tau \boldsymbol{x}_\tau$$

$$= \sum_{q=1}^{B} \lambda_q \langle \hat{\boldsymbol{\mu}}_\tau, \hat{\boldsymbol{\mu}}_q \rangle \langle y_\tau \boldsymbol{x}_\tau, y_q \boldsymbol{x}_q \rangle$$

$$= \lambda_\tau \|\hat{\boldsymbol{\mu}}_\tau\|^2 \|\boldsymbol{x}_\tau\|^2 + \sum_{q:\, q \neq \tau} \lambda_q \langle \hat{\boldsymbol{\mu}}_\tau, \hat{\boldsymbol{\mu}}_q \rangle \langle y_q \boldsymbol{x}_q, \boldsymbol{x}_\tau y_\tau \rangle.$$

The above equation also holds if we average over $\tau$ i.e.

$$1 \leq \frac{1}{B} \sum_{\tau=1}^{B} \lambda_\tau \|\hat{\boldsymbol{\mu}}_\tau\|^2 \|\boldsymbol{x}_\tau\|^2 + \frac{1}{B} \sum_{\tau=1}^{B} \sum_{q:\, q \neq \tau} \lambda_q \langle \hat{\boldsymbol{\mu}}_\tau, \hat{\boldsymbol{\mu}}_q \rangle \langle y_q \boldsymbol{x}_q, \boldsymbol{x}_\tau y_\tau \rangle$$

$$= \frac{1}{B} \sum_{\tau=1}^{B} \lambda_\tau \|\hat{\boldsymbol{\mu}}_\tau\|^2 \|\boldsymbol{x}_\tau\|^2 + \frac{1}{B} \sum_{q=1}^{B} \lambda_q \sum_{\tau:\tau \neq q} \langle \hat{\boldsymbol{\mu}}_\tau, \hat{\boldsymbol{\mu}}_q \rangle \langle y_q \boldsymbol{x}_q, \boldsymbol{x}_\tau y_\tau \rangle$$

$$\leq \sum_{\tau=1}^{B} \lambda_\tau \frac{\|\hat{\boldsymbol{\mu}}_\tau\|^2 \|\boldsymbol{x}_\tau\|^2}{B} + \sum_{q=1}^{B} \lambda_q \cdot \left| \frac{1}{B} \sum_{\tau:\tau \neq q} \langle \hat{\boldsymbol{\mu}}_\tau, \hat{\boldsymbol{\mu}}_q \rangle \langle y_q \boldsymbol{x}_q, \boldsymbol{x}_\tau y_\tau \rangle \right|$$

$$\leq \frac{2R^4}{c_B k} \sum_{\tau=1}^{B} \lambda_\tau + c \left( \frac{1}{c_B} \vee 1 \right) \frac{R^4}{k} \log^2(B/\delta) \sum_{q=1}^{B} \lambda_q,$$

where the last inequality holds by the second part of Lemma C.3. We can conclude that

$$\sum_{\tau=1}^{B} \lambda_\tau \geq c \cdot \frac{(c_B \wedge 1)}{\log^2(B/\delta)} \cdot \frac{k}{R^4},$$

which proves the first part of the lemma. The last part of the Lemma regarding $\mathrm{tr}(\boldsymbol{P}^\top \boldsymbol{W}\boldsymbol{P})$ following directly from the displayed Equation and Lemma C.3 (last item).

- $\|\boldsymbol{W}\|_F$ **upper bound**. We first derive an upper bound on $\|\boldsymbol{W}\|_F$ by showing the existence of some matrix $\boldsymbol{U}$ (up to some scaling) that satisfies the constraints of the max-margin problem. Frei and Vardi [11] used $\boldsymbol{U} = \boldsymbol{I}_d$, so a natural candidate in our case (where the signal $\boldsymbol{\mu}$ is sampled from some low-dimensional subspace) is the projection matrix $\boldsymbol{P}\boldsymbol{P}^\top$. Let $\boldsymbol{U} := \boldsymbol{P}\boldsymbol{P}^\top$ to be the projection matrix into the columns of $\boldsymbol{P}$. We first derive an upper bound on $\|\boldsymbol{W}\|_F$ by showing that the matrix $\boldsymbol{U}$ (up to some scaling) satisfies the constraints of the max-margin problem (Problem 4). Indeed, by Lemma C.3 (last item), for any $\tau \in [B]$,

$$\hat{\boldsymbol{\mu}}_\tau^\top \boldsymbol{P}\boldsymbol{P}^\top y_\tau \boldsymbol{x}_\tau = \tilde{\Theta}(R^2).$$

Thus the matrix $\boldsymbol{U}/R^2$ separates the training data with a margin of at least $1$ for every sample. Since W is the minimum Frobenius norm matrix which separates all of the training data with margin 1,this implies

$$\|\boldsymbol{W}\|_F \le \left\|\boldsymbol{U}/R^2\right\|_F = \frac{\sqrt{k}}{R^2}. \tag{38}$$

By Remark D.3 the same bound also holds for $\|\boldsymbol{P}^\top \boldsymbol{W} \boldsymbol{P}\|_F$.

- $\|\boldsymbol{W}\|_F$ **upper bound, when number of tasks** $B \le dk$ **.** The above approach does not yield a tight bound when the number of tasks is small. Instead, we use a different approach and define

$$\boldsymbol{U} := \theta \cdot \sum_{q=1}^{B} y_q \hat{\boldsymbol{\mu}}_q \boldsymbol{x}_q^\top. \tag{39}$$

The idea is to choose specific value for $\theta$ and show that for this specific value, $\boldsymbol{U}$ satisfies the constraints of the max-margin problem, then by definition of $\boldsymbol{W}$ (Thm. 2.3) we can upper bound $\|\boldsymbol{W}\|_F$ by $\|\boldsymbol{U}\|_F$. We start with a thechnical calculation that will be usefull later in the proof. Since *good run* holds, by Lemma C.3 (last part), if $B \le dk$, then for any $\tau \in [B]$ we have,

$$\left| \sum_{q:q \ne \tau} \langle \hat{\boldsymbol{\mu}}_\tau, \hat{\boldsymbol{\mu}}_q \rangle \cdot \langle \boldsymbol{x}_q, \boldsymbol{x}_\tau \rangle \right| \le c_0 \cdot \sqrt{B} \cdot \frac{R^2 \sqrt{d}}{\sqrt{k}} \cdot \log^2(B/\delta). \tag{40}$$

Next, we move to show that $\boldsymbol{U}$ (Eq. 39) satisfies the constraints of the max-margin problem. Indeed, for any $\tau \in [B]$,

$$y_\tau \hat{\boldsymbol{\mu}}_\tau^\top \boldsymbol{U} \boldsymbol{x}_\tau = \theta \cdot \|\hat{\boldsymbol{\mu}}_\tau\|^2 \|\boldsymbol{x}_\tau\|^2 + \theta \cdot \sum_{q:q \ne \tau} \langle \hat{\boldsymbol{\mu}}_\tau, \hat{\boldsymbol{\mu}}_q \rangle \langle y_q \boldsymbol{x}_q, y_\tau \boldsymbol{x}_\tau \rangle$$

$$\ge \theta \cdot \|\hat{\boldsymbol{\mu}}_\tau\|^2 \|\boldsymbol{x}_\tau\|^2 - \theta \cdot \left| \sum_{q:q \ne \tau} \langle \hat{\boldsymbol{\mu}}_\tau, \hat{\boldsymbol{\mu}}_q \rangle \langle y_q \boldsymbol{x}_q, y_\tau \boldsymbol{x}_\tau \rangle \right|$$

$$\overset{(i)}{\ge} \theta \cdot \frac{R^2 d}{2} - \theta \cdot c_0 \sqrt{B} \frac{R^2 \sqrt{d}}{\sqrt{k}} \cdot \log^2(B/\delta)$$

$$\overset{(ii)}{\ge} 1. \tag{41}$$

Inequality $(i)$ uses Lemma C.3 and Eq. 40. Inequality $(ii)$ holds by choosing $\theta := 3/(R^2 d)$ and for small enough $B$ i.e. $B \le kd/(C \log^4(B/\delta))$. Thus, the matrix $\boldsymbol{U}$ (with $\theta = 3/(R^2 d)$) separates the training data with margin at least $1$ for every sample. By Lemma

 we have that

$$\|\boldsymbol{U}\|_F^2 = \theta^2 \cdot \sum_{q=1}^{B} \|\hat{\boldsymbol{\mu}}_q\|^2 \|\boldsymbol{x}_q\|^2 + \theta^2 \cdot \sum_{q \neq \ell} \langle \hat{\boldsymbol{\mu}}_q, \hat{\boldsymbol{\mu}}_\ell \rangle \cdot \langle y_q \boldsymbol{x}_q, y_\ell \boldsymbol{x}_\ell \rangle$$

$$\leq \theta^2 \cdot \sum_{q=1}^{B} \|\hat{\boldsymbol{\mu}}_q\|^2 \|\boldsymbol{x}_q\|^2 + \theta^2 \cdot \sum_{q=1}^{B} \left| \sum_{\ell:\ell \neq q} \langle \hat{\boldsymbol{\mu}}_q, \hat{\boldsymbol{\mu}}_\ell \rangle \cdot \langle y_q \boldsymbol{x}_q, y_\ell \boldsymbol{x}_\ell \rangle \right|$$

$$\overset{(i)}{\leq} \left( \frac{9}{R^4 d^2} \right) B \left( \frac{3R^2 d}{2} \right) - \left( \frac{9}{R^4 d^2} \right) B \cdot \left( c_0 \sqrt{B} \cdot \left( \frac{R^2 \sqrt{d}}{\sqrt{k}} \right) \cdot \log^2(B/\delta) \right)$$

$$\overset{(ii)}{\leq} \left( \frac{9}{R^4 d^2} \right) B \cdot \left( 3R^2 d \right). \tag{42}$$

Inequality $(i)$ uses Lemma C.3, Eq. 40 and $\theta = 3/(R^2 d)$. Inequality $(ii)$ holds for small enough $B$ i.e. whenever

$$d \geq 2c_0 \sqrt{B}(\sqrt{d}/\sqrt{k}) \log^2(B/\delta) \implies B \leq \frac{2c_0 dk}{\log^4(B/\delta)}.$$

Since $\boldsymbol{W}$ is the minimum Frobenius norm matrix which separates all of the training data with margin 1, and together with Eq. 42 we have that

$$\|\boldsymbol{W}\|_F \leq \|\boldsymbol{U}\|_F \leq \frac{6\sqrt{B}}{\sqrt{d}R} = 6\sqrt{c_B} \frac{\sqrt{k}}{R^2}. \tag{43}$$

Combine with Eq. 38, this proves the upper bound of $\|\boldsymbol{W}\|_F$. By Remark D.3 the same bound also holds for $\|\boldsymbol{P}^\top \boldsymbol{W} \boldsymbol{P}\|_F$.

$\square$

## C.4 Concentration inequalities of quadratic forms

In this section, we derive concentration inequalities for quadratic forms, which can be viewed as variants of the Hanson–Wright inequality, tailored to the following terms:

- $\boldsymbol{\mu}^\top \boldsymbol{W} \boldsymbol{\mu}$, where $\boldsymbol{\mu} = \boldsymbol{P} \boldsymbol{\mu}'$ for some matrix $\boldsymbol{P} \in \mathbb{R}^{d \times k}$ and $\boldsymbol{\mu}' \sim \mathsf{Unif}(\tilde{R} \cdot \mathbb{S}^{k-1})$.
- $\boldsymbol{\mu}^\top \boldsymbol{W} \boldsymbol{z}$, where $\boldsymbol{z} \sim \mathsf{N}(\boldsymbol{0}, \boldsymbol{I}_d)$.
- $\boldsymbol{z}^\top \boldsymbol{W} \boldsymbol{z}$, where $\boldsymbol{z}' \sim \mathsf{N}(\boldsymbol{0}, \boldsymbol{I}_d)$.

$\boldsymbol{\mu}^\top \boldsymbol{W} \boldsymbol{\mu}$:

**Lemma C.5** (Hanson-Wright for uniform on the sphere of a subspace). *Let $\tilde{R} > 0$ and $\boldsymbol{W} \in \tilde{\mathbb{R}}^{d \times d}$ be a matrix. If $\boldsymbol{\mu} = \boldsymbol{P} \boldsymbol{\mu}'$ for some matrix $\boldsymbol{P} \in \mathbb{R}^{d \times k}$ and $\boldsymbol{\mu}' \sim \mathsf{Unif}(\tilde{R} \cdot \mathbb{S}^{k-1})$, then for any $t \geq 0$,*

$$\mathbb{P}\left( \left| \boldsymbol{\mu}^\top \boldsymbol{W} \boldsymbol{\mu} - \frac{\tilde{R}^2}{k} \mathrm{tr}(\boldsymbol{P}^\top \boldsymbol{W} \boldsymbol{P}) \right| \geq t \right) = \mathbb{P}\left( \left| \boldsymbol{\mu}^\top \boldsymbol{W} \boldsymbol{\mu} - \mathbb{E}[\boldsymbol{\mu}^\top \boldsymbol{W} \boldsymbol{\mu}] \right| \geq t \right)$$

$$\leq 2 \exp \left( -c \min \left( \frac{t^2 k^2}{\tilde{R}^4 \|\boldsymbol{P}^\top \boldsymbol{W} \boldsymbol{P}\|_F^2}, \frac{tk}{\tilde{R}^2 \|\boldsymbol{P}^\top \boldsymbol{W} \boldsymbol{P}\|_2} \right) \right)$$

$$\leq 2 \exp \left( -c \min \left( \frac{t^2 k^2}{\tilde{R}^4 \|\boldsymbol{P}^\top \boldsymbol{W} \boldsymbol{P}\|_F^2}, \frac{tk}{\tilde{R}^2 \|\boldsymbol{P}^\top \boldsymbol{W} \boldsymbol{P}\|_F} \right) \right),$$

where $c > 0$ is some constant. We also can conclude that

$$\mathbb{P}\left( \left| \boldsymbol{\mu}^\top \boldsymbol{W} \boldsymbol{\mu} - \frac{\tilde{R}^2}{k} \mathrm{tr}(\boldsymbol{P}^\top \boldsymbol{W} \boldsymbol{P}) \right| \geq t \right) \leq 2 \exp \left( -\frac{ctk}{\tilde{R}^2 \|\boldsymbol{P}^\top \boldsymbol{W} \boldsymbol{P}\|_F} \right)$$

.

*Proof.* Write $\boldsymbol{Q} := \boldsymbol{P}^\top \boldsymbol{W} \boldsymbol{P}$ and observe that $\boldsymbol{\mu}^\top \boldsymbol{W} \boldsymbol{\mu} = \boldsymbol{\mu}'^\top \boldsymbol{Q} \boldsymbol{\mu}'$ for $\boldsymbol{\mu}' \sim \mathsf{Unif}(\tilde{R} \cdot \mathbb{S}^{k-1})$. Then the first inequality follows directly from Lemma C.2 in Frei and Vardi [11]. The second inequality holds since $\|\boldsymbol{W}\|_2 \leq \|\boldsymbol{W}\|_F$ for any matrix $\boldsymbol{W}$. Regarding the last part, write $a := td/\tilde{R}^2 \|\boldsymbol{P}^\top \boldsymbol{W} \boldsymbol{P}\|_F$, if $a \geq 1$, we have that $\exp(-\min(a, a^2)) \leq \exp(-a)$. If $a \leq 1$, the above bound still holds for small enough $c$ and since $\mathbb{P}(\cdot) \leq 1$ is a trivial inequality. $\qquad\square$

## $\boldsymbol{\mu}^\top \boldsymbol{W} \boldsymbol{z}$:

**Lemma C.6.** *Let $\boldsymbol{\mu} = \boldsymbol{P}\boldsymbol{\mu}'$ for some semi-orthogonal matrix $\boldsymbol{P} \in \mathbb{R}^{d \times k}$ and $\boldsymbol{\mu}' \sim \mathsf{Unif}(\tilde{R} \cdot \mathbb{S}^{k-1})$. Moreover, let $\boldsymbol{g} \sim \mathsf{N}(\boldsymbol{0}, \boldsymbol{I}_d)$ (independent of $\boldsymbol{\mu}'$) and let $\boldsymbol{W} \in \mathbb{R}^{d \times d}$ be a matrix. There is an absolute constant $c > 0$ such that for any $t \geq 0$,*

$$\mathbb{P}\left(|\boldsymbol{\mu}^\top \boldsymbol{W} \boldsymbol{g}| \geq t\right) \leq 2 \exp\left(-\frac{ct\sqrt{k}}{\tilde{R}\|\boldsymbol{W}\|_F}\right).$$

*Proof.* By Remark D.1, $\boldsymbol{\mu}$ has sub-Gaussian norm at most $cR/\sqrt{k}$ for some constant $c > 0$. From this point, the proof follows similarly to the proof of Lemma C.4 from Frei and Vardi [11], and we provide it here for completeness. Observe that $\|\boldsymbol{g}\|_{\psi_2} \leq c$, for some absolute constant $c > 0$. By Vershynin [46, Lemma 6.2.3], this implies that for independent $\boldsymbol{g}_1, \boldsymbol{g}_2 \sim \mathsf{N}(\boldsymbol{0}, \boldsymbol{I}_d)$ and any $\beta \in \mathbb{R}$,

$$\mathbb{E} \exp\left(\beta \frac{\sqrt{k}}{\tilde{R}} \boldsymbol{\mu}^\top \boldsymbol{W} \boldsymbol{g}\right) \leq \mathbb{E} \exp\left(c_1 \beta \boldsymbol{g}_1^\top \boldsymbol{W} \boldsymbol{g}_2\right).$$

Then using the moment-generating function of Gaussian chaos [46, Lemma 6.2.2], for $\beta$ satisfying $|\beta| \leq c_2/\|\boldsymbol{W}\|_2$ we have

$$\mathbb{E} \exp\left(\beta \frac{\sqrt{k}}{\tilde{R}} \boldsymbol{\mu}^\top \boldsymbol{W} \boldsymbol{g}\right) \leq \mathbb{E} \exp\left(c_1 \beta \boldsymbol{g}_1^\top \boldsymbol{W} \boldsymbol{g}_2\right)$$
$$\leq \exp(c_3 \beta^2 \|\boldsymbol{W}\|_F^2).$$

That is, the random variable $\tilde{R}^{-1}\sqrt{k}\boldsymbol{\mu}^\top \boldsymbol{W} \boldsymbol{g}$ is mean-zero and has sub-exponential norm at most $\max(c_2^{-1}, c_3)\|\boldsymbol{W}\|_F$. There is therefore a constant $c_4 > 0$ such that for any $u \geq 0$,

$$\mathbb{P}(|\tilde{R}^{-1}\sqrt{k}\boldsymbol{\mu}^\top \boldsymbol{W} \boldsymbol{g}| \geq u) = \mathbb{P}\left(|\boldsymbol{\mu}^\top \boldsymbol{W} \boldsymbol{g}| \geq \frac{\tilde{R}u}{\sqrt{k}}\right) \leq 2 \exp(-cu/\|\boldsymbol{W}\|_F).$$

Setting $u = t\sqrt{k}/\tilde{R}$ we get

$$\mathbb{P}\left(|\boldsymbol{\mu}^\top \boldsymbol{W} \boldsymbol{g}| \geq t\right) \leq 2 \exp\left(-\frac{ct\sqrt{k}}{\tilde{R}\|\boldsymbol{W}\|_F}\right)$$

$\qquad\qquad\qquad\qquad\qquad\qquad\qquad\qquad\qquad\qquad\qquad\qquad\qquad\qquad\qquad\square$

## $\boldsymbol{z}^\top \boldsymbol{W} \boldsymbol{z}$:

**Lemma C.7** (Lemma C5 from Frei and Vardi [11]). *Let $\boldsymbol{\zeta}, \boldsymbol{\zeta}' \overset{i.i.d.}{\sim} \mathsf{N}(\boldsymbol{0}, \boldsymbol{I}_d)$ and let $\boldsymbol{W} \in \mathbb{R}^{d \times d}$ be a matrix. There is a constant $c > 0$ such that for all $t \geq 0$,*

$$\mathbb{P}(|\boldsymbol{\zeta}^\top \boldsymbol{W} \boldsymbol{\zeta}'| \geq t) \leq 2 \exp\left(-\frac{ct}{\|\boldsymbol{W}\|_F}\right).$$

### C.5   Proof of Thm. 3.1

For notational simplicity let us denote $\boldsymbol{W} = \boldsymbol{W}_{\mathrm{MM}}$, $\hat{\boldsymbol{\mu}} := \frac{1}{M}\sum_{i=1}^{M} y_i \boldsymbol{x}_i$, and let us drop the $M+1$ subscript so that we denote $(\boldsymbol{x}_{M+1}, y_{M+1}) = (\boldsymbol{x}, y)$. We recall that $B = c_B \cdot dk/R^2$ (Assumption C.1), and let's denote $\rho$ as the quantity

$$\rho := c \cdot \frac{(c_B \wedge 1)}{\log^2(B/\delta)} \in (0, 1).$$

Then the test error is given by the probability of the event,

$$\{\text{sign}(\hat{y}(E; W)) \neq y_{M+1}\} = \{\hat{\boldsymbol{\mu}}^\top \boldsymbol{W} y \boldsymbol{x} \leq 0\}.$$

By standard properties of the Gaussian, we have for $\boldsymbol{z}, \boldsymbol{z}' \overset{\text{i.i.d.}}{\sim} \mathsf{N}(\boldsymbol{0}, \boldsymbol{I}_d)$,

$$\hat{\boldsymbol{\mu}} \overset{\text{d}}{=} \boldsymbol{\mu} + M^{-1/2} \boldsymbol{z},$$

$$\tilde{y} \boldsymbol{x} \overset{\text{d}}{=} \boldsymbol{\mu} + \boldsymbol{z}'.$$

We thus have

$$
\begin{aligned}
&\mathbb{P}(\hat{y}(E; W) \neq y_{M+1}) \\
&= \mathbb{P}\Big((\boldsymbol{\mu} + M^{-1/2}\boldsymbol{z})^\top \boldsymbol{W}(\boldsymbol{\mu} + \boldsymbol{z}') < 0\Big) \\
&= \mathbb{P}\Big(\boldsymbol{\mu}^\top \boldsymbol{W} \boldsymbol{\mu} < -\boldsymbol{\mu}^\top \boldsymbol{W} \boldsymbol{z}' - M^{-1/2} \boldsymbol{z}^\top \boldsymbol{W} \boldsymbol{\mu} - M^{-1/2} \boldsymbol{z}^\top \boldsymbol{W} \boldsymbol{z}'\Big). \\
&\leq \mathbb{P}\Big(\boldsymbol{\mu}^\top \boldsymbol{W} \boldsymbol{\mu} < \big|\boldsymbol{\mu}^\top \boldsymbol{W} \boldsymbol{z}'\big| + \big|M^{-1/2} \boldsymbol{z}^\top \boldsymbol{W} \boldsymbol{\mu}\big| + \big|M^{-1/2} \boldsymbol{z}^\top \boldsymbol{W} \boldsymbol{z}'\big|\Big) \\
&\leq \mathbb{P}\left(\boldsymbol{\mu}^\top \boldsymbol{W} \boldsymbol{\mu} < \frac{\rho \tilde{R}^2}{2R^2} \cup \big|\boldsymbol{\mu}^\top \boldsymbol{W} \boldsymbol{z}'\big| \geq \frac{\rho \tilde{R}^2}{8R^2} \cup \big|M^{-1/2} \boldsymbol{z}^\top \boldsymbol{W} \boldsymbol{\mu}\big| \geq \frac{\rho \tilde{R}^2}{8R^2} \cup \big|M^{-1/2} \boldsymbol{z}^\top \boldsymbol{W} \boldsymbol{z}'\big| \geq \frac{\rho \tilde{R}^2}{8R^2}\right) \\
&\leq \mathbb{P}\left(\boldsymbol{\mu}^\top \boldsymbol{W} \boldsymbol{\mu} < \frac{\rho \tilde{R}^2}{2R^2}\right) + \mathbb{P}\left(\big|\boldsymbol{\mu}^\top \boldsymbol{W} \boldsymbol{z}'\big| \geq \frac{\rho \tilde{R}^2}{8R^2}\right) + \mathbb{P}\left(\big|M^{-1/2} \boldsymbol{z}^\top \boldsymbol{W} \boldsymbol{\mu}\big| \geq \frac{\rho \tilde{R}^2}{8R^2}\right) \\
&\quad + \mathbb{P}\left(\big|M^{-1/2} \boldsymbol{z}^\top \boldsymbol{W} \boldsymbol{z}'\big| \geq \frac{\rho \tilde{R}^2}{8R^2}\right),
\end{aligned}
\tag{44}
$$

where the last two inequalities hold since given events $\mathbb{B}_1 \subseteq \mathbb{B}_2$ we have that $\mathbb{P}(\mathbb{B}_1) \leq \mathbb{P}(\mathbb{B}_2)$ and by the union bound. Next, we assume that *good run* occurs, which indeed happened with probability at least $1 - 5\delta$ over the draws of the training set (see Definition C.1). We now proceed by bounding each of the remaining terms in the inequality above:

- $\mathbb{P}\Big(\boldsymbol{\mu}^\top \boldsymbol{W} \boldsymbol{\mu} < \frac{\rho \tilde{R}^2}{2R^2}\Big)$. For the first term we can use Lemma C.5 with $t = \frac{\rho \tilde{R}^2}{2R^2}$ to obtain

$$
\begin{aligned}
\mathbb{P}\left(\boldsymbol{\mu}^\top \boldsymbol{W} \boldsymbol{\mu} \leq \frac{\tilde{R}^2}{k} \text{tr}(\boldsymbol{P}^\top \boldsymbol{W} \boldsymbol{P}) - t\right) &= \mathbb{P}\left(\boldsymbol{\mu}^\top \boldsymbol{W} \boldsymbol{\mu} \leq \frac{\tilde{R}^2}{k} \text{tr}(\boldsymbol{P}^\top \boldsymbol{W} \boldsymbol{P}) - \frac{\rho \tilde{R}^2}{2R^2}\right) \\
&\leq 2\exp\left(-\frac{ck}{\tilde{R}^2 \|\boldsymbol{P}^\top \boldsymbol{W} \boldsymbol{P}\|_F} \cdot \frac{\rho \tilde{R}^2}{2R^2}\right).
\end{aligned}
\tag{45}
$$

In Lemma C.4, we show that $\text{tr}(\boldsymbol{P}^\top \boldsymbol{W} \boldsymbol{P}) \geq \frac{\rho k}{R^2}$. By substituting that into the displayed equation, we obtain

$$
\begin{aligned}
\mathbb{P}\left(\boldsymbol{\mu}^\top \boldsymbol{W} \boldsymbol{\mu} \leq \frac{\rho \tilde{R}^2}{2R^2}\right) &\leq 2\exp\left(-\frac{ck}{\tilde{R}^2 \|\boldsymbol{P}^\top \boldsymbol{W} \boldsymbol{P}\|_F} \cdot \frac{\rho \tilde{R}^2}{2R^2}\right) \\
&\overset{(i)}{\leq} 2\exp\left(-\frac{ck}{\tilde{R}^2} \cdot \frac{R^2}{(1 \wedge \sqrt{c_B})\sqrt{k}} \cdot \frac{\rho \tilde{R}^2}{2R^2}\right) \\
&= 2\exp\left(-c \cdot \frac{\rho \sqrt{k}}{1 \wedge \sqrt{c_B}}\right).
\end{aligned}
$$

Inequality $(i)$ uses the upper bound $\|\boldsymbol{P}^\top \boldsymbol{W} \boldsymbol{P}\|_F \leq (1 \wedge \sqrt{c_B}) \cdot \frac{\sqrt{k}}{R^2}$ from Lemma. C.4.

- $\mathbb{P}\left(\left|\boldsymbol{\mu}^\top \boldsymbol{W} \boldsymbol{z}'\right| \geq \frac{\rho \tilde{R}^2}{8R^2}\right)$ **and** $\mathbb{P}\left(\left|M^{-1/2}\boldsymbol{z}^\top \boldsymbol{W} \boldsymbol{\mu}\right| \geq \frac{\rho \tilde{R}^2}{8R^2}\right)$. The second and the third term in Eq. 44 can be bounded by Lemma C.6. Indeed,

$$
\mathbb{P}\left(\left|\boldsymbol{\mu}^\top \boldsymbol{W} \boldsymbol{z}'\right| \geq \frac{\rho \tilde{R}^2}{8R^2}\right) \leq 2\exp\left(-\frac{c\sqrt{k}}{\tilde{R}\|\boldsymbol{W}\|_F} \cdot \frac{\rho \tilde{R}^2}{8R^2}\right)
$$

$$
\overset{(i)}{\leq} 2\exp\left(-\frac{c\sqrt{k}}{\tilde{R}} \cdot \frac{R^2}{(1 \wedge \sqrt{c_B})\sqrt{k}} \cdot \frac{\rho \tilde{R}^2}{8R^2}\right)
$$

$$
= 2\exp\left(-\frac{c\rho\tilde{R}}{8(1 \wedge \sqrt{c_B})}\right). \tag{46}
$$

Inequality $(i)$ uses the upper bound $\|\boldsymbol{W}\|_F \leq (1 \wedge \sqrt{c_B}) \cdot \frac{\sqrt{k}}{R^2}$ from Lemma. C.4.

- $\mathbb{P}\left(\left|M^{-1/2}\boldsymbol{z}^\top \boldsymbol{W} \boldsymbol{z}'\right| \geq \frac{\rho \tilde{R}^2}{8\sqrt{kd}}\right)$. For the final term in Eq. 44 we can use Lemma C.7,

$$
\mathbb{P}\left(\left|M^{-1/2}\boldsymbol{z}^\top \boldsymbol{W} \boldsymbol{z}'\right| \geq \frac{\rho \tilde{R}^2}{8R^2}\right) = \mathbb{P}\left(\left|\boldsymbol{z}^\top \boldsymbol{W} \boldsymbol{z}'\right| \geq \frac{\rho M^{1/2} \tilde{R}^2}{8R^2}\right)
$$

$$
\leq 2\exp\left(-\frac{c}{\|\boldsymbol{W}\|_F} \cdot \frac{\rho M^{1/2} \tilde{R}^2}{8R^2}\right)
$$

$$
\overset{(i)}{\leq} 2\exp\left(-\frac{cR^2}{(1 \wedge \sqrt{c_B})\sqrt{k}} \cdot \frac{\rho M^{1/2} \tilde{R}^2}{8R^2}\right)
$$

$$
= 2\exp\left(-\frac{c\rho M^{1/2} \tilde{R}^2}{(1 \wedge \sqrt{c_B})\sqrt{k}}\right). \tag{47}
$$

Again Inequality $(i)$ uses the upper bound of $\|\boldsymbol{P}^\top \boldsymbol{W} \boldsymbol{P}\|_F$ from Lemma. C.4.

Putting together Eqs. 45, 46 and 47 we get

$$
\mathbb{P}(\hat{y}(E;W) \neq y_{M+1}) \leq 2\exp\left(-c \cdot \frac{\rho\sqrt{k}}{1 \wedge \sqrt{c_B}}\right) + 2\exp\left(-\frac{c\rho\tilde{R}}{8(1 \wedge \sqrt{c_B})}\right) + 2\exp\left(-\frac{c\rho M^{1/2} \tilde{R}^2}{(1 \wedge \sqrt{c_B})\sqrt{k}}\right)
$$

By plugging $\rho = c \cdot \frac{(c_B \wedge 1)}{\log^2(B/\delta)}$, we can upper bound the displayed equation by:

$$
6\exp\left(-\frac{c}{\log^2(B/\delta)} \cdot (1 \wedge \sqrt{c_B}) \cdot \left(\sqrt{k} \wedge \tilde{R} \wedge \frac{M^{1/2}\tilde{R}^2}{\sqrt{k}}\right)\right)
$$

Recall that $B = c_B \cdot dk/R^2$ (Assumption C.1), which means that $c_B = BR^2/dk$. We can conclude that

$$
\mathbb{P}(\hat{y}(E;W) \neq y_{M+1}) \leq 6\exp\left(-\frac{c}{\log^2(B/\delta)} \cdot \left(1 \wedge \sqrt{\frac{BR^2}{dk}}\right) \cdot \left(\sqrt{k} \wedge \tilde{R} \wedge \frac{M^{1/2}\tilde{R}^2}{\sqrt{k}}\right)\right).
$$

# D  Additional Lemmas

**Remark D.1.** *Let $\boldsymbol{P} \in \mathbb{R}^{d \times k}$ be an semi-orthogonal matrix i.e. $\boldsymbol{P}^\top \boldsymbol{P} = \boldsymbol{I}_k$. Let $\boldsymbol{\mu}' \sim \mathsf{Unif}(R \cdot \mathbb{S}^{k-1})$ and set $\boldsymbol{\mu} = \boldsymbol{P}\boldsymbol{\mu}'$. Then $\boldsymbol{\mu}$ and $\boldsymbol{\mu}'$ are sub-Gaussian random vectors with sub-Gaussian norm at most $cR/\sqrt{k}$ for some absolute constant $c > 0$ i.e. $\|\boldsymbol{\mu}\|_{\psi_2} \wedge \|\boldsymbol{\mu}'\|_{\psi_2} \leq cR/\sqrt{k}$.*

*Proof.* $\boldsymbol{\mu}'$ is a sub-Gaussian random vector with sub-Gaussian norm at most $cR/\sqrt{k}$ [46, Theorem 3.4.6] for some absolute constant $c > 0$. In particular, we have that [46, Definition 3.4.1]

$$
\|\boldsymbol{\mu}\|_{\psi_2}^2 = \sup_{\boldsymbol{x} \in \mathbb{S}^{d-1}} \|\langle \boldsymbol{\mu}, x\rangle\|_{\psi_2}^2 = \sup_{\boldsymbol{x} \in \mathbb{S}^{d-1}} \|\langle \boldsymbol{\mu}', \boldsymbol{P}^\top x\rangle\|_{\psi_2}^2
$$

$$
\leq \sup_{\boldsymbol{x} \in \mathbb{S}^{k-1}} \|\langle \boldsymbol{\mu}', x\rangle\|_{\psi_2}^2 = \|\boldsymbol{\mu}'\|_{\psi_2}^2,
$$

where the second equality holds since $\|\boldsymbol{P}^\top \boldsymbol{x}\| \leq \|\boldsymbol{x}\|$ for any $\boldsymbol{x} \in \mathbb{R}^d$. $\qquad \square$

**Remark D.2.** *Let $\boldsymbol{P} \in \mathbb{R}^{d \times k}$ be an semi-orthogonal matrix (i.e. $\boldsymbol{P}^\top \boldsymbol{P} = \boldsymbol{I}_k$) and let $\boldsymbol{z} \sim N(\boldsymbol{0}, \boldsymbol{I}_d)$, then $\boldsymbol{P}^\top \boldsymbol{z} \sim N(\boldsymbol{0}, \boldsymbol{I}_k)$.*

*Proof.*

$$\mathbb{E}[\boldsymbol{P}^\top \boldsymbol{z}] = \boldsymbol{P}^\top \mathbb{E}[\boldsymbol{z}] = \boldsymbol{0}$$
$$Cov(\boldsymbol{P}^\top \boldsymbol{z}) = \mathbb{E}[(\boldsymbol{P}^\top \boldsymbol{z})(\boldsymbol{P}^\top \boldsymbol{z})^\top] = \boldsymbol{P}^\top \mathbb{E}[\boldsymbol{z}\boldsymbol{z}^\top]\boldsymbol{P} = \boldsymbol{P}^\top \boldsymbol{P} = \boldsymbol{I}_k$$

$\square$

**Lemma D.1** (Hoeffding's theorem). *Let $x_1, x_2, \ldots, x_n$ be independent random variables such that $\mathbb{E}[x_i] = 0$ and $x_i \in [-a, a]$ almost surely. Consider the sum of these random variable $S_n = x_1 + \cdots + x_n$. Then Hoeffding's theorem states that*

$$\Pr[|S_n| \geq n^{0.75}a] \leq 2\exp(-2n^{1.5}a^2/4a^2n) = 2\exp(-n^{0.5}/2),$$

*Moreover, if $n \geq 4\log(2/\delta)^2$, then*

$$\Pr[|S_n| \geq n^{0.75}a] \leq \delta,$$

**Lemma D.2.** *Let $\boldsymbol{W} = \sum_{q=1}^B y_q \hat{\boldsymbol{\mu}} \boldsymbol{x}^\top$. Then*

$$\|\boldsymbol{W}\|_F^2 = \sum_{q=1}^B \|\hat{\boldsymbol{\mu}}_q\|^2 \|\boldsymbol{x}_q\|^2 + \sum_{q \neq \ell} \langle \hat{\boldsymbol{\mu}}_q, \hat{\boldsymbol{\mu}}_\ell \rangle \cdot \langle y_q \boldsymbol{x}_q, y_\ell \boldsymbol{x}_\ell \rangle$$

*Proof.*

$$W_{i,j} = \sum_{q=1}^B y_q (\hat{\boldsymbol{\mu}}_q \boldsymbol{x}_q^\top)_{i,j}$$

$$\|\boldsymbol{W}\|_F^2 = \sum_{i,j} \left( \sum_{q=1}^B y_q (\hat{\boldsymbol{\mu}}_q \boldsymbol{x}_q^\top)_{i,j} \right)^2 = \sum_{i,j} \left( \sum_{q,\ell} y_q (\hat{\boldsymbol{\mu}}_q \boldsymbol{x}_q^\top)_{i,j} y_\ell (\hat{\boldsymbol{\mu}}_\ell \boldsymbol{x}_\ell^\top)_{i,j} \right)$$

$$= \sum_{i,j} \left( \sum_{q=1}^B (\hat{\boldsymbol{\mu}}_q \boldsymbol{x}_q^\top)_{i,j}^2 + \sum_{q \neq \ell} y_q y_\ell (\hat{\boldsymbol{\mu}}_q \boldsymbol{x}_q^\top)_{i,j} (\hat{\boldsymbol{\mu}}_\ell \boldsymbol{x}_\ell^\top)_{i,j} \right)$$

$$= \sum_{q=1}^B \|\hat{\boldsymbol{\mu}}_q \boldsymbol{x}_q^\top\|_F^2 + \sum_{q \neq \ell} y_q y_\ell \langle \hat{\boldsymbol{\mu}}_q \boldsymbol{x}_q^\top, \hat{\boldsymbol{\mu}}_\ell \boldsymbol{x}_\ell^\top \rangle_F$$

Recall that $\|A\|_F^2 = tr(A^\top A), \langle A, B \rangle_F = tr(A^T B), tr(A^\top B) = tr(BA^\top)$, which means that $\|\hat{\boldsymbol{\mu}}_q \boldsymbol{x}_q^\top\|_F^2 = \|\hat{\boldsymbol{\mu}}_q\|^2 \|\boldsymbol{x}_q\|^2$ and $\langle \hat{\boldsymbol{\mu}}_q \boldsymbol{x}_q^\top, \hat{\boldsymbol{\mu}}_\ell \boldsymbol{x}_\ell^\top \rangle_F = \langle \hat{\boldsymbol{\mu}}_q, \hat{\boldsymbol{\mu}}_\ell \rangle \cdot \langle \boldsymbol{x}_q, \boldsymbol{x}_\ell \rangle$. Substituting that into the displayed equation gives us the desired result. $\square$

**Remark D.3.** *Let $\boldsymbol{P} \in \mathbb{R}^{d \times k}$ be an semi-orthogonal matrix and let $\boldsymbol{W} \in \mathbb{R}^d$ be a matrix. Then*

$$\|\boldsymbol{P}^\top \boldsymbol{W} \boldsymbol{P}\|_F \leq \|\boldsymbol{W}\|_F$$

*Proof.* Given 2 matrices $A \in \mathbb{R}^{n \times m}, \boldsymbol{B} \in \mathbb{R}^{m \times d}$ it is will known that $\|\boldsymbol{A}\boldsymbol{B}\|_F \leq \|\boldsymbol{A}\| \|\boldsymbol{B}\|_F$. Therefore,

$$\|\boldsymbol{P}^\top \boldsymbol{W} \boldsymbol{P}\|_F \leq \|\boldsymbol{W}\boldsymbol{P}\|_F = \|\boldsymbol{P}^\top \boldsymbol{W}^\top\|_F \leq \|\boldsymbol{W}^\top\|_F = \|\boldsymbol{W}\|_F$$

$\square$

# E Further Experiments & Additional Details

In this section, we provide further experiments and additional details. We trained linear attention models (Eq. (3)) on data generated as specified in Section 2.1 using GD with a fixed step size $\alpha = 0.01$, $N = 40$ and the logistic loss function. Training was performed for $200 - 300$ steps from a zero initialization, implemented in PyTorch. In all figures, the x-axis represents the number of in-context examples $M$, starting from $M = 1$, while the y-axis corresponds to the test accuracy. The accuracy is computed by checking whether the prediction $\hat{y}(\boldsymbol{E}; \boldsymbol{W}_t)$ equals the true label $y_{M+1}$, where $\boldsymbol{E}$ denotes the tokenization of the sequence $(\boldsymbol{x}_1, y_1), \ldots, (\boldsymbol{x}_M, y_M), (\boldsymbol{x}_{M+1}, 0)$, used to predict $y_{M+1}$. We average the accuracies over $B_{\text{test}} := 1200$ tasks, and we plot this average, optionally including error bars representing $95\%$ confidence intervals (CIs). These intervals are computed using the standard error of the mean and assuming a normal distribution:

$$CI = [\overline{acc} - z\hat{\sigma}, \overline{acc} + z\hat{\sigma}],$$

where $\overline{acc} := \sum_i a\hat{c}c_i / B_{\text{test}}$ is the accuracy mean, $\hat{\sigma} := \sqrt{\sum_i (a\hat{c}c_i - \overline{acc})^2 / (B_{\text{test}} - 1)}$ is the standard error of the mean and $z = 1.96$. All computations can be completed within an hour on a CPU.

In Figure 2, we examine the effect of the shared subspace dimension $k$, with each plot corresponding to a different signal strength. Accuracy improves as the signal strength $R = \tilde{R}$ increases and as the subspace dimension $k$ decreases. Notably, in some plots, the accuracy remains far from 1 even for large values of $M$. This can be attributed to the term $\sqrt{k} \wedge \tilde{R} \wedge M\tilde{R}^4/k$ that appears in the generalization bound of Theorem 3.1. Indeed, when $M$ is sufficiently large, the dominant term becomes $\sqrt{k} \wedge \tilde{R}$, indicating that perfect accuracy (i.e., zero error) is unattainable in this regime.

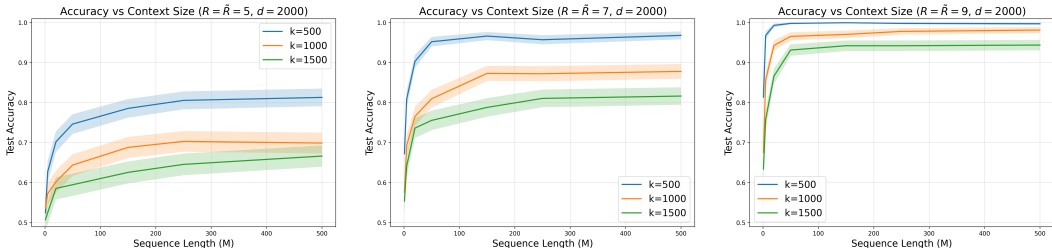

Figure 2: Test accuracy versus the number of in-context examples $M$ for various subspace dimensions $k$. Accuracy improves as the signal strength $R = \tilde{R}$ increases and as the subspace dimension $k$ decreases. $B = d = 2000$.

