# OpenReview forum: "Transformers are almost optimal metalearners for linear classification"
_NeurIPS.cc/2025/Conference — NeurIPS 2025 poster_

### Official Review · Reviewer_uuw9 · 2025-06-30

**Clarity:** 2
**Significance:** 3
**Originality:** 3
**Rating:** 3
**Confidence:** 4

**Summary:**

This paper theoretically investigates that  ICL with transformer can match the optimal meta-learner, on linear regression tasks with class-conditional Gaussian mixture model, where all tasks share the same input space.

**Questions:**

Please refer to weakness.

**Ethical Concerns:**

["NO or VERY MINOR ethics concerns only"]

**Final Justification:**

After rebuttal, my second concern was addressed, so I raised my score. But my first concern remains critical, and seems not could be addressed by discussion or minor revision.

**Quality:**

2

**Strengths And Weaknesses:**

The main conclusion in this paper that with sufficient training, ICL with transformer acts like optimal meta-learners, is reasonable.
However, it seems that [1] also gives such result, in more general settings, empirically without theoretical guarantee. But [1] is not mentioned at all.

Though the reviewer generally agrees with the main conclusion, there are some critical concerns:
1. It seems that the results have nothing to do with ICL with transformer. It seems that without Sec 2.2, by replacing it with any model parameterized by W, the sequel still holds. As the sequel only rely on the setting/assumptions, rather then the model.
2. The formal result (Theorem 3.1) is confusing. The review understand the way to decomposition of model into task-specific $f$ and shared $h$, which is common in multitask-learning and meta-learning [2]. In the particular setting (Sec 2.1) where the problem is linear and $h$ is linear (the step 1&2 in assumption 2.1/2.2 is no other than replacing the input from uniform distribution on unit sphere in $R^k$ with $R^d$), as far as the reviewer could understand, useful information can only been brought when d<k, as the noise $z$ would has less dimensions with fixed $I_d$. However, the main result (theorem 3.1) shows that the meta-learning risk would increase with d decreases, which is contradictory.

The reviewer would consider to raise the score if related works and above concerns can be discussed properly.

[1] Wu S, Wang Y, Yao Q. Why In-Context Learning Models are Good Few-Shot Learners?[C]//The Thirteenth International Conference on Learning Representations.
[2] Maurer A, Pontil M, Romera-Paredes B. The benefit of multitask representation learning[J]. Journal of Machine Learning Research, 2016, 17(81): 1-32.

---

> ### Author Rebuttal · Authors · 2025-07-30
>
> We thank the reviewer for their thoughtful feedback. Below we respond to each concern and question in order.
>
> **Missing Citation to [1] (Wu et al.):**  Wu et al. (2025) present an empirical study suggesting that transformers can approximate optimal meta-learners in few-shot classification tasks. While their results are encouraging, the work is primarily empirical and does not offer formal guarantees. The only theoretical result in the paper is Theorem 2.1, which establishes an expressivity claim, that is, the existence of transformer weights that can simulate a metalearning algorithm. However, they do not analyze whether such a solution can be obtained through optimization, which is a central focus of our work.
>
> In addition, their analysis relies on the universal approximation property of multi-layer perceptrons (MLPs), without specifying the required network size. In contrast, our model does not include an MLP component. Finally, it is unclear whether the algorithms studied in their paper achieve optimality in our Gaussian mixture setting.
>
> Our contribution complements this line of work by providing the first formal guarantee that a simple transformer trained by gradient descent can achieve near-optimal meta-learning performance under well-defined assumptions.
>
> We agree that Wu et al. is a relevant and timely reference, and we will cite their work in the revised version.
>
>
> **W1. Is the Result Specific to Transformers?**
>
> The answer is yes. While our theoretical guarantee in Theorem 3.1 is expressed in terms of the max-margin solution $W_{mm}$ , this solution is directly tied to the specific transformer architecture presented in Eq. 3. Section 2.2 introduces a linear attention-based transformer model, which has been studied in several prior works, and our analysis closely follows the behavior of this model when trained via gradient descent.
>
> Theorem 2.3 defines the max-margin solution $W_{mm}$ in terms of this attention model, as noted in line 215. Specifically, the constraint in Eq. (4) reflects the prediction structure of the model described in Eq. (3). **If a different model were used, the constraints would differ, leading to a different optimization problem and, in general, a different solution.** As a result, our analysis would no longer apply in that case.
>
> **W2. Clarification on Theorem 3.1 and the Role of d and k:**
>
> We believe there may be a misunderstanding:
>
> **Clarification of the setting:** In our setting, each task's signal vector $\mu$ lies in a $k$-dimensional subspace of $\mathbb{R}^d$, and the noise is isotropic in $\mathbb{R}^d$. Therefore, **the data is only defined for $k \leq d$** (see line 162). Indeed, Assumption 2.1 requires that the matrix $P \in R^{d \times k}$, whose columns span the shared subspace, is semi-orthogonal. That is, $P^TP=I_k$, which is only possible when $k \leq d$, as stated in the assumption.
>
> Each task's signal $\mu \in \mathbb{R}^d$ is generated by embedding a vector $\mu' \sim \text{Unif}(R \cdot \mathbb{S}^{k-1})$ into $R^d$ via $\mu = P \mu'$. Importantly, **$\mu$ is not uniformly distributed over the unit sphere in $\mathbb{R}^d$**; with probability 1, it lies entirely within the shared subspace and has no component in directions orthogonal to it.
>
> In standard metalearning terminology, the optimal shared representation in our setting is given by $h: \mathbb{R}^d \rightarrow \mathbb{R}^k$, where $h(x) = P^T x$. For a signal vector $\mu = P\mu’$, this gives: $h(\mu) = P^T\mu = P^TP\mu’ = \mu’$, which recovers the original signal in the k-dimensional subspace.
>
> Given a task with signal vector $\mu \in \mathbb{R}^d$, constructed from  $\mu’ \in \mathbb{R}^k$, the task-specific classifier $f: \mathbb{R}^k \rightarrow (-1, +1)$ is defined as $f(x) = \text{sign}(\mu'^T x)$.
>
> **Risk would increase with d decreases.**  Our result in Theorem 3.1 shows that the meta-learning error decreases as the ambient dimension $d$ decreases, not the other way around. In particular, as $d$ becomes smaller, the SNR during training increases, which facilitates the learning process and leads to better generalization..

---

> > ### Comment · Reviewer_uuw9 · 2025-08-04
> > **Thanks for your response**
> >
> > Thanks for the authors' response. The rebuttal has addressed my second concern, which was raised by my neglecting the important semi-orthogonality of $P$ and minus sign in  Theorem 3.1.
> > Thus I would raise my rating.
> >
> > However, the first concern remains critical. Note that the whole paper is bridged with ICL with transformer by Theorem 2.3, which is considered to be a relative general solution, for a wide range of models simplified enough.

---

> > > ### Author Response · Authors · 2025-08-04
> > >
> > > We clarify that Theorem 2.3 explicitly characterizes the maximum-margin solution (precisely the solutions to which gradient descent converges) for the transformer-based attention model defined concretely in Eq. (3). The left-hand side of the constraints in Theorem 2.3 corresponds exactly to the model in Eq. (3) where it is applied to the training tasks. Thus, the scope and applicability of our results are explicitly tied to the specific model we study, rather than to a broad or overly simplified abstraction.
> > >
> > > Importantly, the model we analyze in Eq. (3) is **identical** to the model previously studied in multiple prominent theoretical analyses of in-context learning, particularly for linear classification and linear regression ([1-4], all appearing in top-tier ML conferences). Furthermore, other variants of single-layer linear-attention models have been widely used in the literature to facilitate theoretical analysis of in-context learning, as shown in [5–8] and many other studies. Thus, our model from Eq. (3) has become a common test bed for theoretical analyses of in-context learning in transformers.
> > >
> > > Given this extensive precedent, we believe that our choice of model is well-justified. Therefore, we respectfully argue that our model choice does not constitute a critical limitation; rather, it places our work firmly within a well-established research direction.
> > >
> > >
> > >
> > > [1] Juno Kim, Tai Nakamaki, and Taiji Suzuki. Transformers are minimax optimal nonparametric in-context learners. NeurIPS 2024.
> > >
> > > [2] Jingfeng Wu, Difan Zou, Zixiang Chen, Vladimir Braverman, Quanquan Gu, and Peter L Bartlett. How many pretraining tasks are needed for in-context learning of linear regression? ICLR 2024.
> > >
> > > [3] Spencer Frei and Gal Vardi. Trained transformer classifiers generalize and exhibit benign overfitting in-context. ICLR 2025.
> > >
> > > [4] Wei Shen, Ruida Zhou, Jing Yang, and Cong Shen. On the training convergence of transformers for in-context classification. ICML 25.
> > >
> > > [5] Ruiqi Zhang, Spencer Frei, Peter L. Bartlett . Trained Transformers Learn Linear Models In-Context. JMLR 2024.
> > >
> > > [6] Oswald  ET AL. Transformers Learn In-Context by Gradient Descent. ICML 23.
> > >
> > > [7] Ahn ET AL. Transformers learn to implement preconditioned gradient descent for in-context learning. NeurIPS 23.
> > >
> > > [8] Mahankali et al. One step of gradient descent is provably the optimal in-context learner with one layer of linear self-attention.

---

> > > ### Author Response · Authors · 2025-08-07
> > >
> > > We would like to confirm whether our response addressed the reviewer's concern. If not, could the reviewer please clarify whether the concern is about Theorem 2.3 or the model presented in Eq. (3)?

---

### Official Review · Reviewer_ibKw · 2025-07-03

**Clarity:** 3
**Significance:** 3
**Originality:** 3
**Rating:** 4
**Confidence:** 3

**Summary:**

This paper analyzes in-context learning (ICL) with a single-layer linear attention model on Gaussian mixture tasks. It provides theoretical results showing that such an architecture is an almost optimal metalearner for binary classification. The analysis carefully treats the finite-sample setting, which makes it distinct from many prior theoretical ICL works that assume a population training loss.

**Questions:**

1. Do Remarks 2.1 and 2.2 rely on Assumption A? If so, could the authors clarify this connection?
2. A recent paper shows that a single-layer linear attention layer can indeed learn the optimal binary classifier. Could the authors clarify it?
 - Li, Yingcong, et al. When and How Unlabeled Data Provably Improve In-Context Learning. arXiv:2506.15329 (2025).
3. What specific model architecture is used in the experiments? Are the authors implementing Eq. (2) by replacing softmax with linear attention, or are they directly simulating Eq. (3)?
4. The experimental curves are quite noisy. It would be helpful to run more trials and report the averaged results with confidence intervals to ensure the trends are robust.
5. Have the authors tried comparing to the simplest baseline: estimating the task mean directly using $\sum_iy_ix_i$? How does its performance compare to the linear attention model?

**Ethical Concerns:**

["NO or VERY MINOR ethics concerns only"]

**Final Justification:**

The authors have addressed most of my concerns. I suggest incorporating more discussion on comparison to the Bayes error. I have decided to raise my score.

**Limitations:**

yes

**Quality:**

3

**Strengths And Weaknesses:**

Strengths:
1. The paper is clearly written and provides rigorous results in a finite-sample regime, addressing a realistic scenario where the number of tasks and samples per task is finite, which is an important distinction from many previous ICL theory works that assume infinite training data.

2. The main results, particularly Theorem 3.1, are accompanied by detailed discussion and intuition, helping the reader understand the key contributions and their implications.

Weaknesses:
1. Theorem 2.3 holds under the assumption that the data is linearly separable so that an SVM solution exists. In real in-context learning, demonstration samples are typically sampled randomly, which may not always yield separable sets. The paper would benefit from clarifying how $W_{MM}$  in Theorem 2.3 relates to practical training, as well as the feasibility  of $W_{MM}$ for Theorem 3.1.
2. The result in Theorem 2.3 align with optimizing a single weight matrix $W$ as in Eq. (3). However, the actual attention architecture (Eq.(2)) is more complex. it involves separable $W_q$, $W_k$ matrices, which can lead to different optimization behaviors. Prior work has shown that separately optimizing $W_q$ and $W_k$ can implicitly minimize the unclear norm of $U$ in Eq. (4) rather than Frobenius norm. Thus, the paper's theoretical focus on Eq. (3) does not fully capture the behavior of the true Transformer attention mechanism.
3. The paper lacks a concrete comparison between the bound in Theorem 3.1 and the Bayes optimal loss. Additionally, the Remarks 2.1 and 2.2 are not consistent with Theorem 3.1 since they do not appear to account for Assumption A, which seems critical for making the comparison meaningful. It would be better to show the main results as the excess risk between the "single-task optimal learner" and "optimal learner with access to the ground-truth representation". Additionally, simply showing that each error is constant in [0,1] is less informative. It would be more compelling to show whether the error decays exponentially with increasing samples, context length or dimension.

---

> ### Author Rebuttal · Authors · 2025-07-30
>
> We thank the reviewer for their thoughtful feedback. Below we respond to each concern and question in order.
>
> **W1:**
> - **Feasibility of $W_{MM}$:**  We note that this is explicitly addressed in our analysis. See Lemma C.4 and the proof sketch (line 325) for details, where we show a specific construction (i.e. the projection matrix) that separates the data (with high probability over the data distribution). While extending the analysis to non-separable data is an important direction for future work, we believe that studying the separable setting is a meaningful and informative starting point.
>
> - **Data Setting Justification:** We believe our assumptions are reasonable for a theoretical work that gives a precise analysis of the effects of all of the core ingredients to the transformer training pipeline: the implicit regularization effect of gradient descent in pre-training, the number of pre-training tasks, the number of examples per pre-training task, the signal-to-noise ratio of the pre-training tasks, the number of in-context examples at test time and the signal-to-noise ratio of the test-time tasks. Moreover, Gaussian mixture models is a well-studied setting in the literature, which enables us to derive a tight lower bound on the performance of an optimal metalearner (see Remark 2.2). Perhaps surprisingly, our analysis shows that the trained transformer nearly matches this optimal performance. We also note that Frei and Vardi [3] studied in-context learning in transformers using a very similar setting. To the best of our knowledge, there are currently no theoretical works on transformer-based in-context learning for classification that use significantly more realistic or complex datasets and provide such comprehensive results.
>
> [3] Spencer Frei and Gal Vardi. Trained transformer classifiers generalize and exhibit benign overfitting in-context. ICLR 2025.
>
> - **Relation between $W_{MM}$ and practical training:** Formally, the model is guaranteed to converge to $W_{MM}$ asymptotically as the time $t$ tends to infinity.  In our experiments, we find that gradient descent trained for a finite number of steps already achieves strong generalization, closely matching the behavior predicted by the max-margin solution. We note that analyzing the asymptotic solution rather than a finite-time solution is common in many theoretical papers.
>
> **W2:** We agree that analyzing a more general transformer architecture with separate key and query matrices is an important and interesting direction for future work. In our work, we adopt the simplified attention model with a merged key-query matrix, which is commonly used in theoretical studies of transformers and in-context learning (e.g. [3-7], and many more). This setting allows a tractable analysis of the optimization dynamics and implicit bias.
>
> Regarding the reviewer’s comment on nuclear norm minimization, we believe this refers to [2]. As we mentioned in the paper, their analysis relies on strong and somewhat artificial assumptions, such as the existence of a single “optimal” token, and all other tokens receiving identical “scores”. These assumptions are not satisfied in our setting and, in fact, are violated by almost all data distributions with more than two tokens drawn randomly.
>
> While our model does not capture the full complexity of attention with separate key and query matrices, we believe it serves as a meaningful and interpretable proxy that sheds light on how gradient descent interacts with attention in metalearning tasks. Also, as we already mentioned, this is a common setting in theoretical works on transformers.
>
> [2] Davoud Ataee Tarzanagh, Yingcong Li, Christos Thrampoulidis, and Samet Oymak. Transformers as support vector machines.
>
> [3] Johannes Von Oswald, Eyvind Niklasson, Ettore Randazzo, João Sacramento, Alexander Mordvintsev, Andrey Zhmoginov, and Max Vladymyrov. Transformers learn in-context by gradient descent. ICML 23.
>
> [4] Kwangjun Ahn, Xiang Cheng, Hadi Daneshmand, and Suvrit Sra. Transformers learn to implement preconditioned gradient descent for in-context learning. NeurIPS 23.
>
> [5] Ruiqi Zhang, Spencer Frei, and Peter L Bartlett. Trained transformers learn linear models in-context. JMLR 2024.
>
> [6] Davoud Ataee Tarzanagh, Yingcong Li, Xuechen Zhang, and Samet Oymak. Max-margin token selection in the attention mechanism.
>
> [7] Vasudeva, Bhavya, Puneesh Deora, and Christos Thrampoulidis. ”Implicit bias and fast convergence rates for self-attention.”
>
> **W3:**
>
>  **Error is not a constant** : We are not sure why the reviewer mentioned that the error in our result is constant. Theorem 3.1 shows that the error decays exponentially when several parameters increase simultaneously, specifically, when the subspace dimension k, the test-time signal strength \Tilda(R), and the quantity \sqrt{M\Tilde(R)^4/k}  all increase, assuming the number of training tasks B is large enough to recover the shared representation.
>
> **Comparison with Bayes optimal:**
> - In lines 260-268, we compare the transformer against a Bayes optimal algorithm that has access just to the samples from the new task. While the latter requires \Omega(d/\tilde{R^4}) examples to achieve small constant error (d is the ambient dimension and \Tilde(R) is the signal strength during test), the transformer needs only O(k/\tilde{R^4}) in-context examples, as long the number of tasks is large enough.
>
> - In lines 269-275 we compare the transformer against a Bayes optimal algorithm that also has full access to the shared sub-space, which is a very strong baseline, since metalearners don’t have access to the sub-space, and they need to learn it during training. We showed that both have the same in-context sample complexity, up to polylogarithmic factors.
>
> **Relation between Assumption A to Remark 2.1 and 2.2:**  We clarify that Remarks 2.1 and 2.2 do not rely on Assumption A. These remarks present information-theoretic lower bounds that hold independently of any specific algorithm or training setup. In contrast, Assumption A is used specifically to prove our main result (Theorem 3.1) and ensures that the max-margin solution generalizes with high probability.
>
> As discussed in lines 248--253:
>
> - Assumption A1: requires that the signal strength $R$ is at least $\widetilde{\Theta}(1)$. This is the interesting regime, since Remarks 2.1 and 2.2 show that learning becomes impossible when the test-time signal $\widetilde{R}$ is $o(1)$.
>
>  - Assumption A2 requires that the ambient dimension $d$ is at least polylogarithmic in the number of tasks. This modest assumption, while not needed in Remarks 2.1 and 2.2, is standard in high-dimensional analysis and ensures concentration bounds for sub-exponential random variables.
>
> - Assumption A3 requires the number of examples per task during training to be at least $N \geq d / R^2$, or equivalently $1/\text{SNR}^2$. This is necessary for learning the shared representation during pretraining. However, Remarks 2.1 and 2.2 analyze test-time learners that either do not have access to the shared sub-space $P$ (Remark 2.1) or have oracle access to $P$ (Remark 2.2), so their bounds are independent of $N$.
>
> **Q1.** See the previous part i.e. Relation between Assumption A to Remark 2.1 and 2.2.
>
> **Q2.**
> **Comparison to Li et al.:** We thank the reviewer for pointing us to this relevant reference. While the setting in Li et al. differs from ours in important ways, it is still conceptually related and warrants citation in our paper.
>
> - First, their work does not address the metalearning setting. In particular, the tasks in their framework do not share a common representation (e.g., a low-dimensional subspace), which makes their setting fundamentally different and not directly comparable to ours. In fact, their focus is closer to semi-supervised learning and naturally their bounds depend on the ambient dimension $d$, while our work avoids that.
>
> - Second, they analyze the squared loss, which is more natural for regression and allows for closed-form solutions. In contrast, we study the logistic loss, which is standard for classification and does not admit a closed-form solution. To handle this, we leverage the implicit bias of GD to characterize the behavior of the trained model.
>
> - Third, their results do not clearly describe how key parameters influence generalization. For example, how the number of tasks during training affects the generalization error. In our work, we provide explicit relationships between generalization error and quantities such as the number of training tasks, the number of samples per task, the ambient dimension, shared subspace dimension, and the SNR both in training and test.
>
>
>
> **Q3.** We are directly simulating Eq. (3)
>
> **Q4.** We have included this in the experimental section of the appendix (see the supplementary materials, last section).
>
> **Q5.** Yes, this is precisely the Bayes optimal algorithm, that is, the maximum likelihood estimator (MLE) that we use as a baseline in our experiments (see the discussion at the beginning of Section 5 in the main paper, as well as the experiments presented both in the main text and in the appendix). We evaluate two versions of MLE: one that only uses the average of the in-context test examples, and another that has access to the ground-truth subspace matrix P, which first projects the data using P and then applies MLE. Our results show that the linear transformer significantly outperforms the MLE baseline that lacks access to P, and closely matches the performance of the MLE with projection.

---

> > ### Comment · Reviewer_ibKw · 2025-08-05
> >
> > Thank you for your response.
> >
> > Given the radius $R$ and noise level 1, the Bayes error is $\Phi(-R)$, where $\Phi$ is the cdf of the standard normal. It is a constant. With infinite data, all effective classification algorithms are expected to converge to this bound. Therefore, it would be more meaningful to study the gap to the Bayes error rather than the raw classification error.
> >
> > Overall, the authors have addressed most of my concerns. I suggest incorporating a comparison to the Bayes error in both the theoretical discussion and the experiments. I have decided to raise my score.

---

> > > ### Author Response · Authors · 2025-08-07
> > >
> > > We thank the reviewer for their valuable feedback and suggestions. While our current comparison focuses on a finite number of in-context examples, in the final version, we will include a comparison with the asymptotic case, where the number of in-context examples tends to infinity.

---

### Official Review · Reviewer_5r5s · 2025-07-03

**Clarity:** 3
**Significance:** 4
**Originality:** 4
**Rating:** 6
**Confidence:** 4

**Summary:**

The paper considers the ICL ability of linear transformers on a Gaussian mixture of linear classification.
It shows that a trained transformer is nearly an optimal metalearner.
The result also shows the number of pretraining tasks needed to generalize at test time, where the number is not dependent on the ambient dimension.

**Questions:**

I have a single question on the validity of the data generation setting, which will affect whether my score is accept:

In Assumption 2.2 (and similarly in Assumption 2.1), is it true that the $x_i$ with $y_i=1$ is of the same distribution as the $x_i$ with $y_i=-1$? Since $\mu'$ is uniform on the sphere and $P$ is isometric, it looks to me that the answer is yes. In that case, how is classification possible?

This question is particularly intriguing to me since in Line 261 and Line 272, the paper is claiming that the algorithm can achieve "an arbitrarily small constant" test error. I am not exactly sure how it is possible in the current setting, and would be eager to hear from the authors.

**Ethical Concerns:**

["NO or VERY MINOR ethics concerns only"]

**Final Justification:**

Thank the authors for their rebuttal. It addresses my concern. I believe this work is quite important for the community since it theoretically demonstrates a very important and strong practical feature (meta-learning) of a very influential model (linear transformer). I will raise the score from 4 to 6 and recommend this paper for wider exposure, like oral or spotlight, depending on how the venue is organized.

I recommend that the authors somehow reflect their response into the paper in Assumptions 2.1, 2.2, and Ln 186-188. It looks to me that my initial misunderstanding is quite natural, since it is currently not explicitly mentioned in Assumptions 2.1 and 2.2 that they are the *task* distribution rather than the *data* distribution. For example:

Assumption 2.1' (training-time **task** distribution). ... For any task $\tau \in [B]$, the input and label pairs $(x_{\tau,i}, y_{\tau,i})_{i=1}^{N+1}$ in task $\tau$ satisfies the following:
1. Let $\mu'_\tau$ be sampled from a prior distribution $\text{Unif}(\tilde{R}\cdot S^{k-1})$, i.e., the uniform distribution on the sphere of radius $\tilde{R}\$ in $k$ dimensions.
...

This and something similar for Assumption 2 should work. Also, in Ln 186, you might want to specify that "In our setting, each example is a task $(x_i, y_i)_{i=1}^{N+1}$ that consists of a sequence of $(x,y)$ pairs".

I believe something like these edits can prevent other people from suffering the same misunderstanding as mine. This can also keep the notation consistent with Ln 257 in Theorem 3.1.

Please feel free to comment if this suggestion makes sense.

Also, I disagree with Reviewer uuw9's evaluation on Eq. (3). As mentioned in my review, it is a standard tool that has been present in prominent papers in this field.

**Limitations:**

N/A on societal impact.

**Paper Formatting Concerns:**

Some minor typos:
- In Line 305, "Gausian"

**Quality:**

4

**Strengths And Weaknesses:**

This is a technical solid paper that studies a rather important problem in ICL theory.
It considers the mixture of classification tasks, which is an important theoretical setting that I do not think has been properly addressed in previous work.
The theoretical result is informative and demonstrates the ICL ability of linear transformer in such setting.

---
I will detail my observation as follows.

- Methods And Evaluation Criteria:

The data distribution setting, namely the Gaussian mixture of linear classification task, is a rather novel setting and an important step toward understanding the ICL ability of linear transformers.
ICL on single-distribution linear regularization has been investigated in e.g. [1], and a potential setting on the Gaussian mixture of linear regularization has been proposed in a recent ICML submission [2].
The Gaussian mixture of linear classification task is a step forward.
Since it is a novel setting, I would like to further check with the authors on whether the setting is reasonable, i.e., it does not refer to some degenerate case or some cases where metalearning is unachievable.
The definition itself in Section 2.1 is mathematically rigid, but I would like to seek some clarification on its implication.
Therefore, I will specify my questions on the problem setting in later sections.

- Theoretical Claims:

I skimmed through the proofs including the proof sketch in Section 4 and the Appendix.
The proof uses several standard tools in analyzing linear transformer, namely Eq. (3) which has been present in [1, 2].
The proof involves some standard concentration lemmas, and the concentration lemmas are rather established in the textbooks, so it looks to me that the proved result stands as it is.

The main result reduces the bound on in-context examples so that it no longer depends on the ambient dimension $d$, which is a good improvement.

- Experimental Designs Or Analyses:

The numerical experiment replicates and justifies the theoretical findings.
It is not the main contribution in this paper anyway, so it does not affect my evaluation.


- Relation To Broader Scientific Literature:
It is closely related to the theoretical in-context learning literature. It studies the in-context learning ability on a Gaussian mixture of linear classification models, which is a step forward from the non-mixed linear regression/classification function class that is present in the literature.

If the mixture of linear classification setting makes sense, this is surely a good paper for this venue. However, I have some doubts about the setting, as stated before. Therefore, I have a single question on the setting which will greatly impact whether my score is accept or reject, which is listed in the questions below.

---

[1] Zhang, R., Wu, J., & Bartlett, P. (2024). In-context learning of a linear transformer block: Benefits of the mlp component and one-step gd initialization. Advances in Neural Information Processing Systems, 37, 18310-18361.

[2] Jin, Y., Balasubramanian, K., & Lai, L. (2024). In-context Learning for Mixture of Linear Regressions: Existence, Generalization and Training Dynamics. arXiv preprint arXiv:2410.14183.

---

> ### Author Rebuttal · Authors · 2025-07-30
>
> We thank the reviewer for their thoughtful feedback.
> Regarding your question, we believe you are referring to Assumptions 2.1 and 2.2 (there is no Assumption 3.1 or 3.2). The answer is no,  but it is essential to also condition on $\mu$. That is, the distributions of $x_i$ conditioned on $y_i = 1$ and $y_i = -1$ are not the same, when we also condition on $\mu$.
>
> In both assumptions, the task-specific vector $\mu$ is sampled once for each task and fixed. Then,
>
> $x_i = y_i \cdot \mu + z_i,$
>
> where $z_i \sim \mathcal{N}(0, I_d)$. Thus, conditioned on $\mu$, the positive and negative classes are Gaussian distributions centered at $\mu$ and $-\mu$, respectively. This separation makes classification possible despite the symmetry in the prior over $\mu$. Thus, since our in-context sequence consists of many examples $(x_i,y_i)$ which are drawn from a Gaussian mixture that corresponds to the same $\mu$, then given enough in-context examples we can correctly classify fresh examples which are drawn using this $\mu$.
>
> We hope this clarifies the data generation process and addresses the concern about class distinguishability.

---

> ### Comment · Reviewer_5r5s · 2025-08-06
>
> Thank the authors for their rebuttal. It addresses my concern. I believe this work is quite important for the community since it theoretically demonstrates a very important and strong practical feature (meta-learning) of a very influential model (linear transformer). Also, it investigates a Gaussian mixture of logistic regression setting, which is the first in the ICL theory field. This is a theoretical technique advance. I will raise the score from 4 to 6 and recommend this paper for wider exposure, like oral or spotlight, depending on how the venue is organized.
>
> I recommend that the authors somehow reflect their response into the paper in Assumptions 2.1, 2.2, and Ln 186-188. It looks to me that my initial misunderstanding is quite natural, since it is currently not explicitly mentioned in Assumptions 2.1 and 2.2 that they are the *task* distribution rather than the *data* distribution. For example:
>
> Assumption 2.1' (training-time **task** distribution). ... For any task $\tau \in [B]$, the input and label pairs $(x_{\tau,i}, y_{\tau,i})_{i=1}^{N+1}$ in task $\tau$ satisfies the following:
> 1. Let $\mu'_\tau$ be sampled from a prior distribution $\text{Unif}(\tilde{R}\cdot S^{k-1})$, i.e., the uniform distribution on the sphere of radius $\tilde{R}\$ in $k$ dimensions.
> ...
>
> This and something similar for Assumption 2.2 should work. Also, in Ln 186, you might want to specify that "In our setting, each example is a task $(x_i, y_i)_{i=1}^{N+1}$ that consists of a sequence of $(x,y)$ pairs".
>
> I believe something like these edits can prevent other people from suffering the same misunderstanding as mine. This can also keep the notation consistent with Ln 257 in Theorem 3.1.
>
> Please feel free to comment if this suggestion makes sense.
>
> Also, I disagree with Reviewer uuw9's evaluation on Eq. (3). As mentioned in my review, it is a standard tool that has been present in prominent papers in this field.

---

> > ### Author Response · Authors · 2025-08-07
> >
> > We thank the reviewer for the positive feedback and are glad our rebuttal addressed the concern. We agree that the distinction between task distributions and data distributions within the same task should be made clearer. We will revise Assumptions 2.1, 2.2, and Line 186 in the final version to reflect this, following the reviewer’s helpful suggestions.

---

> ### Comment · Reviewer_5r5s · 2025-08-06
>
> For Assumptions 2.1 and 2.2, one can also follow the writing of Assumption 1 in [1].
>
> [1] Jingfeng Wu, Difan Zou, Zixiang Chen, Vladimir Braverman, Quanquan Gu, and Peter L Bartlett. How many pretraining tasks are needed for in-context learning of linear regression? ICLR 2024.

---

### Official Review · Reviewer_PK2N · 2025-07-23

**Clarity:** 3
**Significance:** 2
**Originality:** 3
**Rating:** 3
**Confidence:** 4

**Summary:**

This paper studied the in-context learning of one-layer linear self-attention models to do binary classification on a Gaussian mixture dataset. After training such a transformer, it has learned the ground truth projection, so it only requires fewer test samples to achieve a small test error. It reduces the sample complexity compared with the optimal binary classifier on the test dataset. This result improved previous results of [11] by Spencer Frei and Gal Vardi. This paper, employing a straightforward linear model, theoretically demonstrated that transformers can function as metalearners, capable of adapting to novel tasks using only a limited number of in-context examples, without the need for additional training.

**Questions:**

1. Line 179: there should not be index $\tau$. Line 305: Gausian

2. In section 2.2, I understand the Tokenization is normal in many theory paper, but can we consider the embedding different from (1)? For instance, $\mathbf{E}=(x_1^\top,y_1,\ldots,x_n^\top,y_n,x_{n+1}^\top,0)$.

3. In Theorem 2.3, if we consider nonlinear softmax attention, can we still get some similar solution for the gradient descent training?

4. In theorem 3.1, what is $\mathbf{E}$? Is it training or test dataset?

5. In the last remark in Section 3, the authors can eliminate the dependence of $B$ and $N$ on $d$ in the analysis. Can you compare such results with the classical PCA method for Gaussian mixture models, e.g. [1]? In [1], the threshold for exact recovery is also governed by sample size and SNR. Also, it is worth mentioning [2] as another direction for studying transformers using Gaussian mixture models.

-----------------------------------------------------------------------------------------------
[1] An $l_p$ theory of PCA and spectral clustering

[2] Transformers as Unsupervised Learning Algorithms: A study on Gaussian Mixtures

**Ethical Concerns:**

["NO or VERY MINOR ethics concerns only"]

**Final Justification:**

The authors partially addressed my concerns. I greatly appreciate their effort and detailed discussion. However, I cannot agree with their first response: Eq. (3) serves as a standard toy model for theoretical analyses of in-context learning, so we can focus on this toy model and do not need to enhance our analysis to make it more realistic for practical transformer models. I believe Eq. (3) does not fully capture the behavior of the genuine Transformer attention mechanism. As for the impact on practitioners, I think this paper uses this toy model to elucidate some ideas that may already be familiar to practitioners. Based on the novelty of the mathematical methods and the final results presented in the paper, I will maintain my current score.

**Limitations:**

The paper should have a detailed discussion of the limitations in the main text. The limitation of the theory may come from the assumption of the Gaussian mixture model and the linear self-attention model.

**Quality:**

2

**Strengths And Weaknesses:**

Strengths: This paper is well-organized and states the theorems in a very clear way. It is technically sound with sufficient discussions on the results and the proof is clear.

Weaknesses: The main concern is the assumptions of the dataset and attention model in the theory. This paper only considers the linearly separable binary classification and a convex parameterization of the linear transform. These may loose many crucial properties of the transformer both in pretraining and test process. Additionally, it should be more impactful if the authors could extend the results to in-context multi-class classification like reference [12]. And although getting theory for more general setting may be hard, the paper could present more general simulations to show its result can be extended to more general settings. Finally, it is not clear what kind of message a machine learning practitioner can obtain from this theoretical work.

---

> ### Author Rebuttal · Authors · 2025-07-30
>
> We thank the reviewer for their feedback. Below, we respond to each point raised.
>
> **Simplified Architecture:** While we analyze a 1-layer linear transformer, this simplification allows us to rigorously characterize fundamental properties of in-context learning that may extend to more complex architectures. Similar to how early theoretical works on neural networks focused on simplified architectures to establish foundational results, our work provides a theoretical framework that can guide future analyses of more complex transformers.
>
> Moreover, our architecture is exactly identical to the architecture considered in at least four prior works on in-context learning in linear classification or linear regression [1-4 below] (all of them appeared in top tier ML conferences), and other variants of 1-layer linear attention models have appeared in many other works on in-context learning in linear regression tasks (5-8, and much more). Thus, this model has become a standard toy model for theoretical analyses of in-context learning, and it clearly attracts much interest in the theoretical deep-learning community. Hence, we feel that recommending rejection based on this reason would be unfair.
>
> [1] Juno Kim, Tai Nakamaki, and Taiji Suzuki. Transformers are minimax optimal nonparametric in-context learners. NeurIPS 2024.
>
> [2] Jingfeng Wu, Difan Zou, Zixiang Chen, Vladimir Braverman, Quanquan Gu, and Peter L Bartlett. How many pretraining tasks are needed for in-context learning of linear regression? ICLR 2024.
>
> [3] Spencer Frei and Gal Vardi. Trained transformer classifiers generalize and exhibit benign overfitting in-context. ICLR 2025.
>
> [4] Wei Shen, Ruida Zhou, Jing Yang, and Cong Shen. On the training convergence of transformers for in-context classification. ICML 25.
>
> [5] Ruiqi Zhang, Spencer Frei, Peter L. Bartlett . Trained Transformers Learn Linear Models In-Context. JMLR 2024.
>
> [6] Oswald  ET AL. Transformers Learn In-Context by Gradient Descent. ICML 23.
>
> [7] Ahn ET AL. Transformers learn to implement preconditioned gradient descent for in-context learning. NeurIPS 23.
>
> [8] Mahankali et al. One step of gradient descent is provably the optimal in-context learner with one layer of linear self-attention.
>
>
> **Data Setting:** We believe our assumptions are reasonable for a theoretical work that gives a precise analysis of the effects of all of the core ingredients to the transformer training pipeline: the implicit regularization effect of gradient descent in pre-training, the number of pre-training tasks, the number of examples per pre-training task, the signal-to-noise ratio of the pre-training tasks, the number of in-context examples at test time and the signal-to-noise ratio of the test-time tasks. Moreover, Gaussian mixture models is a well-studied setting in the literature, which enables us to derive a tight lower bound on the performance of an optimal metalearner (see Remark 2.2). Perhaps surprisingly, our analysis shows that the trained transformer nearly matches this optimal performance. We also note that Frei and Vardi [3] studied in-context learning in transformers using a very similar setting. To the best of our knowledge, there are currently no theoretical works on transformer-based in-context learning for classification that use significantly more realistic or complex datasets and provide such comprehensive results.
>
> [3] Spencer Frei and Gal Vardi. Trained transformer classifiers generalize and exhibit benign overfitting in-context. ICLR 2025.
>
> **Impact for Practitioners**
> We showed that simple linear attention can efficiently encode task structure when the tasks lie in a low-dimensional subspace, and that transformers can approach optimal performance with substantially fewer in-context examples, assuming sufficient pretraining. These insights may guide architectural decisions in real-world few-shot applications. That said, the primary goal of this work, like many theoretical studies, is to deepen our understanding of how transformers perform metalearning and to clarify the relationships between key factors such as the number of pretraining tasks, the number of examples per task, and the SNR to enable metalearning. We are aware that theoretical lines of research often take time to translate into significant practical impact.
>
>
> **Respond for Questions:**
> **Q1:** Thank you for pointing this out. We will fix that in the revised version.
>
>
> **Q2:** We are not entirely sure we understand the proposed tokenization. In our setup, each column of $E$ corresponds to a different token. As a general principle, each token fed into the transformer must have the same dimension. If the idea is to tokenize each $(x_i,y_i)$ pair as two separate tokens, this raises several issues. First, it would require padding $y_i$  to match the dimension of $x_i$, which adds unnecessary complexity. More importantly, this design leads to attention interactions between label tokens and input tokens within the same example, rather than between different examples. We therefore suspect that such a tokenization would impair generalization and increase the in-context sample complexity.
>
> We agree that analyzing different tokenization is an interesting question.
>
>
> **Q3:**  As noted in the discussion of our “simplified architecture”, our analysis relies on the linear attention model, where the implicit bias of gradient descent is well understood. We believe it should be possible to extend our analysis to non-convex linear transformer architectures which are homogeneous in the parameters, since the KKT condition analysis approach can work in that setting as well. Unfortunately, the implicit bias of gradient descent in the softmax attention setting is not yet fully characterized (see references in line 151), which makes it challenging to extend our proof techniques to that case. Developing such a theory for softmax-based transformers remains an important and open direction, as we mentioned in Section 6.
>
>
> **Q4.**  Thank you for pointing this out. $E$ refers to the test data. Specifically, the tokenization of the sequence $(x_1,y_1), \ldots, (x_M,y_M), (x_{M+1}, 0)$, used to predict $y_{M+1}$. We agree that this was not clearly stated, and we will add an explicit clarification in the final version.
>
>
> **Q5:**  In PCA  the data matrix $X \in \mathbb{R}^{n \times d}$ is given, and the goal is to compute a rank-$k$ approximation that minimizes the Frobenius norm error. The rank $k$ is known and fixed, and the objective is purely reconstruction-based.
>
> In contrast, in our setting, the rank $k$ is not provided in advance, and the transformer must implicitly infer a relevant subspace through the attention mechanism in the course of solving a different task: namely, in-context linear classification. That is, the attention head learns to separate labeled data points according to class, rather than recover the underlying data geometry. This difference in objective makes a direct compression from one setting to the other nontrivial.
>
> Moreover, the guarantees in [9] rely on an asymptotic regime, where the number of samples $n \to \infty$ (see Assumption 2.1 therein). Translating to our setting, this would require the number of tasks to tend to infinity. In contrast, our bounds hold for a finite number of tasks and, notably, are independent of the ambient dimension $d$.
>
> We will add a citation to [10], which takes a complementary approach by showing that transformers are capable of performing unsupervised learning in the Gaussian mixture model (GMM) setting.
>
>  [9] An $\ell_p$ theory of PCA and spectral clustering
>
>  [10] Transformers as Unsupervised Learning Algorithms: A study on Gaussian Mixtures

---

### Note · Authors · 2025-08-13

We thank all reviewers for their time and valuable feedback.

Our motivation is to rigorously advance the theoretical understanding of in-context learning (ICL) for classification tasks, particularly in the metalearning setting, where the goal is to solve related tasks more efficiently than solving each task independently. We believe this setting, though highly relevant in practice, especially in large language models, is underexplored in recent theoretical work on transformer-based ICL, despite its deep roots in the classical learning theory literature. Our work contributes to filling this gap.

We actively participated in the rebuttal process and addressed reviewer concerns. However, we did not receive final replies from Reviewer PK2N and Reviewer uuw9, so we are unsure whether our clarifications addressed their concerns.

Their main concerns relate to the simplicity of our model (linear attention), and the binary classification setup based on class-conditional Gaussian mixture data. As we explained in the rebuttal and throughout the paper, these assumptions are consistent with many recent theoretical works in top-tier venues: Our architecture is identical to at least four ICL studies [1-4], and other variants of linear attention models have been used extensively in regression-focused ICL (e.g. [5-8]). Thus, this architecture has become a common test-bed for studying ICL in transformers. Variants of our data have also been adopted in prior ICL theory (e.g. [3]).  For brevity, reference numbers here correspond to those in our final comment to Reviewer uuw9. We thus believe rejection on this basis alone would be unfair.

In summary, we show that gradient-based training of transformers can match the in-context sample complexity of an optimal metalearner, closely approach the performance of optimal learner with access to the shared subspace, and outperform any learner that only observes in-context examples. Our analysis captures how key parameters affect generalization, including the number of pretraining tasks, examples per task, signal-to-noise ratios (both during training and test), and number of in-context examples. Notably, our bounds on required training tasks/examples do not depend on the ambient dimension, and we improve prior results without a shared representation. Our empirical results support the theory.

We hope this message helps clarify our contribution. Thank you for the opportunity to participate in this process.

---

### Decision · Program_Chairs · 2025-09-17

**Decision:**

Accept (poster)

**Comment:**

This submission studies in-context learning in a transformer as meta-learning, for the restricted case of linear classification. The submission presents theoretical contributions that reviewers found well-motivated and interesting. The primary concern raised across reviews centers on the restrictive nature of the theoretical assumptions. While these assumptions may limit the immediate applicability of the results, the reviewers assessed that the motivation and theoretical insights are sufficiently compelling to warrant acceptance.